# Processing methodology for the ITS_LIVE Sentinel-1 ice velocity product

Yang Lei[1], Alex S. Gardner[2], Piyush Agram[1]

[1]Division of Geological and Planetary Science, California Institute of Technology, Pasadena, 91125, USA
[2]Jet Propulsion Laboratory, California Institute of Technology, Pasadena, 91109, USA

*Correspondence to*: Yang Lei (ylei@caltech.edu)

**Abstract.** The NASA MEaSUREs Inter-mission Time Series of Land Ice Velocity and Elevation (ITS_LIVE) project seeks to accelerate understanding of critical glaciers and ice sheet processes by providing researchers with global, low-latency, comprehensive and state-of-the-art records of surface velocities and elevations as observed from space. Here we describe the
image-pair ice velocity product and processing methodology for ESA Sentinel-1 radar data. We demonstrate improvements to the core processing algorithm for dense offset tracking, "*autoRIFT*", that provides finer resolution (120 m instead of the previous 240 m used for Version 1) and higher accuracy (20% to 50% improvement) data products with significantly enhanced computational efficiency (>2 orders of magnitude) when compared to earlier versions and the state-of-the-art "dense ampcor" routine in JPL's ISCE software. In particular, the disparity filter is upgraded for handling finer grid resolution with overlapping
search chip sizes, and the oversampling ratio in the subpixel cross-correlation estimation is adaptively determined for Sentinel-1 data by matching the precision of the measured displacement based on the search chip size used. A novel calibration is applied to the data to correct for Sentinel-1A/B subswath and full-swath dependent geolocation biases caused by systematic issues with the instruments. Sentinel-1's C-band images are affected by variations in the total electron content of the ionosphere that results in large velocity errors in the azimuth (along-track) direction. To reduce these effects, slant-range (line-of-sight or
LOS) velocities are used and accompanied by LOS parameters that support map coordinate (x/y) velocity inversion from ascending and descending slant-range offset measurements, as derived from two image-pairs. After the proposed correction of ionosphere errors, the uncertainties in velocities are reduced by 9-61%. We further validate the ITS_LIVE Version 2 Sentinel-1 image pair products, with 6-year time series composed of thousands of epochs, over three typical test sites covering the globe: the Jakobshavn Isbræ Glacier of Greenland, Pine Island Glacier of Antarctic, and Malaspina Glacier of Alaska. By
comparing with other similar products (PROMICE, FAU, and MEaSUREs Annual Antarctic Ice Velocity Map products), as well as other ITS_LIVE Version 2 products from Landsat-8 and Sentinel-2 data we find an overall variation between products is around 100 m/yr over fast-flowing glacier outlets, where both mean velocity and variation are on the order of km/yr, and increases up to 300-500 m/yr (3-6%) for the fastest Jakobshavn Isbræ Glacier. The velocity magnitude uncertainty of the ITS_LIVE Sentinel-1 products is calculated to be uniformly distributed around 60 m/yr for the three test regions investigated.
The described product and methods comprise the MEaSUREs ITS_LIVE Sentinel-1 Image-Pair Glacier and Ice Sheet Surface Velocities: Version 2 (doi: https://doi.org/10.5067/0506KQLS6512).

## 1 Introduction

As the planet warms in response to increased concentrations of greenhouse gases in the atmosphere, glaciers and ice sheet are losing more mass to the world's oceans, leading to accelerated rates of sea level rise (IPCC AR6). Glaciers and ice sheets are

losing mass through both accelerated melting and solid ice discharge into the ocean via enhanced flow. How glacier and ice sheet flow will change in the future is one of the largest uncertainties in projections of sea level change. Ice flow velocity is a critical constraint for determining ice discharge (Gardner et al., 2018), modelling ice dynamics with numerical simulations (Farinotti et al., 2019), understanding the short- and long-term ice flow dynamic variation (e.g., seasonal and/or tidal; Minchew et al., 2017; Greene et al., 2020) as well as calculating mass balance budget (e.g., Bamber and Rivera, 2007; Minowa et al.,

2021). Tracking of features in repeat satellite imagery provides a vantage point for measuring ice motion over continental scales (Bindschadler and Scambos, 1991; Frolich and Doake, 1998), providing insights into the processes that drive large-scale changes in flow. Surface velocities have been successfully derived from both optical and radar imagery including NASA's Landsat 4/5/6/7/8 (Fahnestock et al., 2016; Gardner et al., 2018), ESA's Sentinel-1 (Nagler et al., 2015; Andersen et al., 2020; Friedl et al., 2021; Solgaard et al., 2021) and Sentinel-2 (Kääb et al., 2016; Nagy and Andreassen, 2019), DLR's

TerraSAR-X and TanDEM-X (Joughin et al., 2010) and through multi-mission synthesis (Miles et al., 2017; Mouginot et al., 2017a; Derkacheva et al., 2020).

Surface velocities can be derived from pairs of radar images using multiple techniques and programs, which have been cross-compared in a comprehensive way by carefully setting up the programs and coordinating with various groups worldwide

(Boncori et al., 2018). In general, four different approaches have been shown to be capable of measuring surface velocities from radar and/or optical satellite data, i.e., Synthetic Aperture Radar Interferometry (InSAR; Goldstein et al., 1993), Multi-Aperture SAR Interferometry (MAI; Bechor and Zebker, 2006), incoherent offset-tracking (amplitude only; Gray et al., 1998), and coherent offset-tracking (amplitude and phase; Joughin, 2002). Among them, the most accurate technique (i.e. cm level for Sentinel-1) is Synthetic Aperture Radar Interferometry (InSAR) that is highly sensitive to measure changes in range from

repeat-pass observations (Joughin et al., 1995; Rignot et al., 1995; Gray et al., 1998; Joughin et al., 1998; Joughin et al., 1999; Michel and Rignot, 1999; Yu et al., 2010; Gourmelen et al., 2011; Mouginot et al., 2019a). MAI is less widely used than the conventional (or differential) InSAR technique, because it is capable of measuring the azimuth velocity component only, which is thus more subject to the large velocity errors in the azimuth (along-track) direction that are due to the ionosphere effects in SAR processing and temporal decorrelation (Boncori et al., 2018). However, interferometry can be problematic over fast-

flowing ice and/or areas where snow accumulation and melting occur due to rapid temporal decorrelation. In contrast, offset tracking (amplitude only; incoherent) or speckle tracking (amplitude and phase; coherent) have been predominantly used in tracking both along-track and Line-of-Sight (LOS) ice motion as it is less sensitive to phase wrapping errors and temporal decorrelation (Fahnestock et al., 1993; Strozzi et al., 2002; de Lange et al., 2007; Strozzi et al., 2008; Nagler et al., 2012; Riveros et al., 2013; Fahnestock et al., 2016; Mouginot et al., 2017a; Kusk et al., 2018). As Boncori et al., 2018 shows, it is

not yet convinced that the coherent offset-tracking method provides essentially more accurate results than the incoherent counterpart by sacrificing half of the efficiency (two times the runtime).

Besides regional attempts for generating ice flow maps such as over high-mountain glaciers (Dehecq et al., 2015; Millan et al., 2019), several satellites derived large-scale operational ice velocity mappings are released annually (Nagler et al., 2015;

Mouginot et al., 2019b; Joughin, 2020a) or more frequently (Joughin, 2020b; Gardner et al., 2019; Solgaard et al., 2021). Among these efforts, the NASA MEaSUREs project Inter-mission Time Series of Land Ice Velocity and Elevation (ITS_LIVE) releases ice velocity products, i.e., 1) image-pair granules (without time averaging), 2) datacubes (time series of image-pair results) and 3) regional mosaics (averaged both spatially and temporally) with global coverage using temporally dense multi-sensor observations from both optical (Landsat 4/5/6/7/8 and Sentinel-2) and SAR (Sentinel-1) satellite data

(Gardner et al., 2018). Other similar products of regional and/or global ice velocities have also been released, such as the Programme for Monitoring of the Greenland Ice Sheet (PROMICE) Ice Velocity product (Solgaard and Kusk, 2021) that releases temporally (24-day) averaged velocity mosaics over Greenland Ice Sheet, as well as the global products released by Friedrich Alexander University (FAU) that include image-pair (without time averaging) products, monthly and annual mosaics (Friedl et al., 2021). Furthermore, both products of PROMICE and FAU are generated purely using Sentinel-1 SAR dataset

via the offset tracking method, similar to our ITS_LIVE Version 2 Sentinel-1 products, which will then be cross-validated later in this work by comparing with PROMICE and FAU in various regions of the globe. The specifics of all three Sentinel-1 based data products (PROMICE, FAU and ITS_LIVE) are summarized in Table 1. As described Lei et al., 2021a, the core processing algorithm of the ITS_LIVE project utilizes the combination of a precise geocoding module "*Geogrid*" and an efficient offset tracking module "*autoRIFT*" (autonomous Repeat Image Feature Tracking), both of which are open source

(https://github.com/nasa-jpl/autoRIFT) and have been integrated to NASA/JPL's InSAR Scientific Computing Environment (ISCE) software (https://github.com/isce-framework/isce2).

**Table 1.** Sentinel-1 based data product specifics of PROMICE, FAU and ITS_LIVE.

| Data Product | Temporal Resolution | Spatial Resolution | Period | Method | Uncertainty | Global/ Regional |
|---|---|---|---|---|---|---|
| PROMICE | 24 day (temporally averaged) | grid spacing 500 m (effective resolution: 800-900 m) | 2016-present | Normalized Cross Correlation | 20-27 m y$^{-1}$ (with in-situ GPS) and 8-12 m y$^{-1}$ (over stable ground) | Greenland only |
| FAU | 6-12 day (no temporal averaging) | grid spacing 200 m (effective resolution: 800-900 m) | 2016-present | Normalized Cross Correlation | 14.6 m y$^{-1}$ (with TerraSAR-X) and 87.6 m y$^{-1}$ (with Landsat-8) | Global |
| ITS_LIVE | 6-12 day (no temporal averaging) | grid spacing 120 m (effective resolution: 240-1920 m) | 2016-present | Normalized Cross Correlation | 50-70 m y$^{-1}$ (over rocks and stationary surfaces) | Global |

In this work we focus on the ITS_LIVE processing of the C-band Sentinel-1A/B radar data. Specifically, we focus on the processing of the Terrain Observation with Progressive Scan (TOPS) mode of the Interferometric Wide Swath (IW) SAR observations (De Zan and Guarnieri, 2006; Prats-Iraola et al., 2015; Miranda et al., 2017; Schubert et al., 2017). This mode enables surface imaging with a 3.7 m ground range and 15.6 m azimuth resolution with a repeat cycle of 6 days between A and B satellites over polar regions and some other key areas of the world. SAR imagery has qualities that are highly valuable

to imaging of polar glaciers and ice sheets as the instrument is not obscured by cloud or limited by solar illumination. In Section 2.1, we provide an overview of the MEaSUREs ITS_LIVE Sentinel-1 Image-Pair Glacier and Ice Sheet Surface Velocities: Version 2, on which the ITS_LIVE global ice velocity mosaics will be based. For illustration purposes, we show

examples of several Sentinel-1 image pairs collected over Greenland (Lei et al., 2021b), and demonstrate several recent improvements to *autoRIFT* providing finer resolution, improved accuracy, and increased processing efficiency. In Section 2.2
we discuss the adoption of a 120 m regular-spaced output grid (previously a 240 m grid was used) with 50% overlap in search chips and a modified Normalized Displacement Coherence (NDC) filter. In Section 2.3 we discuss using a chip-size-dependent subpixel oversampling ratio that improves sub-pixel accuracy while retaining computational efficiency. In Section 2.4 we present a runtime comparison between *autoRIFT* and ISCE software's default dense feature tracking algorithm, "dense ampcor". In Section 3 we discuss approaches to minimizing systematic geolocation biases (Section 3.1) and effects of
ionosphere disturbance (Section 3.2). In Section 4, we further validate the globally available ITS_LIVE Sentinel-1 image pair products over three typical test sites: the Jakobshavn Isbræ Glacier of Greenland (Section 4.1), Pine Island Glacier of Antarctic (Section 4.2), and Malaspina Glacier of Alaska (Section 4.3), by comparing with other similar products (PROMICE, FAU and other MEaSUREs products) as well as other ITS_LIVE Version 2 products from optical sensors (Landsat-8 and Sentinel-2) available in each region. A summary of the work is provided in Section 5.

## 2 ITS_LIVE Sentinel-1 image pair product description and processing chain

In this section we provide a high-level overview of the ITS_LIVE Sentinel-1 image pair product and the core processing algorithms. Both inputs to and outputs of the processing chain are demonstrated for a set of test data collected over Greenland. We then discuss two improvements to the *autoRIFT* processing chain that are exploited for Version 2 processing of the Sentinel-1 data.

### 2.1 Product and methodology overview

#### 2.1.1 Input dataset

ITS_LIVE Sentinel-1 processing chain inputs and outputs are provided in Table 2.

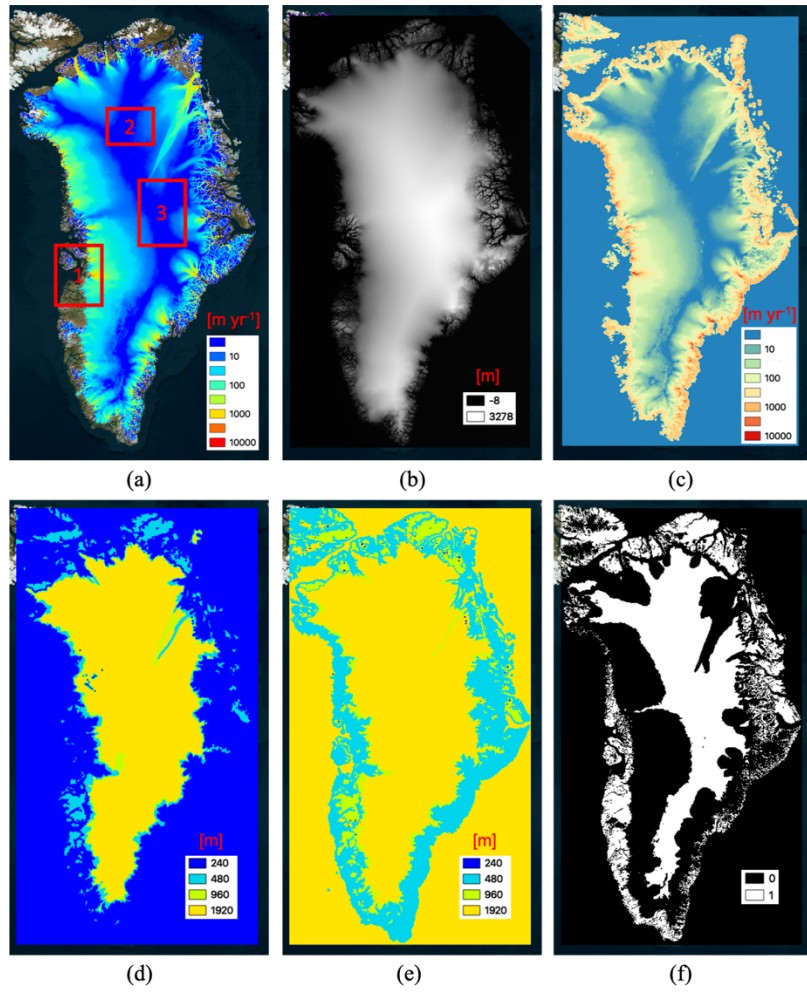

**Figure 1: Input to the Sentinel-1 processing chain: (a) reference velocity magnitude with test sites shown as red rectangles, (b) elevation, (c) search limit in x, (d) minimum chip size, (e) maximum chip size, (f) stable surface mask, that is defined as ice free terrain or areas having a reference velocity < 15 m yr⁻¹.**

**Table 2.** Input and output data for the MEaSUREs ITS_LIVE Sentinel-1 Image-Pair Glacier and Ice Sheet Surface Velocities: Version 2

processing chain. x and y are directions in map coordinates. Each output variable is accompanied with variable specific metadata.

|  | Sentinel-1 image pair |
| --- | --- |
|  | elevation [m above WGS84 ellipsoid] |
|  | topographic slopes in x and y [m m⁻¹] |
| Input | reference velocity in x and y [m yr⁻¹] |
|  | search limit in x and y [m yr⁻¹] |
|  | minimum chip size in x and y [m] |
|  | maximum chip size in x and y [m] |
|  | stable surface mask [binary] |

| | |
|---|---|
| | **vx** [int16]: velocity in x [m yr$^{-1}$] |
| | **vy** [int16]: velocity in y [m yr$^{-1}$] |
| | **v** [int16]: velocity magnitude [m yr$^{-1}$] |
| | **v_error** [int16]: velocity magnitude error [m yr$^{-1}$] |
| | **vr** [int16]: slant range (LOS) velocity [m yr$^{-1}$] |
| | **va** [int16]: azimuth (along-track) velocity [m yr$^{-1}$] |
| Output | **M11** [int16]: velocity conversion matrix element (1$^{st}$ row, 1$^{st}$ column) |
| | **M12** [int16]: velocity conversion matrix element (1$^{st}$ row, 2$^{nd}$ column) |
| | **chip_size_height** [uint16]: height (along azimuth) of chip [m] |
| | **chip_size_width** [uint16]: width (along range) of chip [m] |
| | **interp_mask** [uint8]: interpolation flag, 0 = measured, 1 = interpolated [binary] |
| | **img_pair_info** [char]: image pair metadata |

Example input data for Greenland are illustrated in Fig. 1. In Fig. 1a, we demonstrate the magnitude of reference velocity from the 20-year ice-sheet-wide velocity mosaic (Joughin et al., 2010; Joughin et al., 2016; Joughin et al., 2017) derived from the synthesis of SAR/InSAR data and Landsat-8 optical imagery. In ITS_LIVE processing, both the x- and y-components of the
reference velocity are used to centre the downstream search routine. The three red rectangles in Fig. 1 show the locations used for demonstration and validation of the algorithms. The list of 21 Sentine-1 image pairs used for validation is provided in Table 3 (Lei et al., 2021b). In Fig. 1b, we show the GIMP Digital Elevation Model (DEM) for the Greenland Ice Sheet (Howat et al., 2014; Howat et al., 2015), which is used in this work for illustration purposes only. Note for the global processing, we considered various DEM's with different resolution, e.g. Arctic DEM, REMA DEM, TanDEM-X DEM, Copernicus DEM,
and NASADEM, and found the Copernicus DEM with global coverage at 30 m resolution (GLO-30) is the best available large-scale DEM which was also baselined for NASA-ISRO's NISAR mission after extensive analysis. With this analysis, it is found DEM's with varying resolutions have negligible effects on the resulting offset-tracking velocities given the grid spacing of 120 m used. Geolocation accuracy is however slightly more sensitive to the DEM resolution and accuracy. The DEM, and its derivatives with respect to x- and y-directions, are all used to map between pixel index and displacement in radar range/azimuth
coordinates and geolocation and surface velocity in map-projected Cartesian coordinates (Northings/Eastings). Note with a different DEM, the offset tracking estimates in radar coordinates (slant-range/azimuth) are relatively insensitive to DEM errors, e.g. a DEM change or error of a few tens of meters leads to pixel mis-registration on the order of 0.001 pixels. However, the DEM-derived surface slopes that are used in mapping between radar viewing geometry and the map projected coordinate system are sensitive to the DEM error. Therefore, the map-projected velocity estimates tend to be affected by the DEM slope
error (e.g. over regions with high topographic relief). The search limit is shown in Fig. 1c, which constrains the size of the search window and varies spatially with the reference velocity and proximity to the ocean. A search limit of zero indicates that no offset search should be conducted. Fig. 1d and Fig. 1e are the minimum and maximum allowable chip sizes. *autoRIFT* will cycle from the minimum to the maximum chip size until a "valid" offset is found (i.e., passing the NDC filter based on the derived pixel displacement similarity with adjacent chips; as discussed in Section 2.2), returning the finest achievable

resolution within the specified limits. Here we specify smaller chip sizes for fast-flowing glaciers (lower accuracy but higher spatial resolution) along the margin of Greenland and larger chip sizes (higher accuracy with lower resolution) for the interior regions where gradients in velocity are low (Fig. 1d-e). Fig. 1f shows the stable surface mask that is defined as ice free terrain or areas having a reference velocity < 15 m yr$^{-1}$ and is used for calibration of the velocity fields.

**Table 3.** Sentinel-1 image pairs used for validation of the algorithms. All image pairs were acquired with the Interferometric Wide Swath (IW) Single Look Complex (SLC) mode. Only HH-pol is used. The pixel spacing is 3.67 m in ground range and 15.59 m in azimuth. The parenthesis of "(A/B)" denotes the Sentinel-1A image was acquired prior to the Sentinel-1B image, and vice versa.

| Region 1 (Path: 90 / Frame: 222, 227; ascending) | Region 2 (Path: 31 / Frame: 253; ascending) | Region 3 (Path: 112 / Frame: 350; descending) | Region 3 (Path: 31 / Frame: 233, 238; ascending) |
|---|---|---|---|
| 20170104-20170110 (A/B) | 20171226-20180101 (A/B) | 20161002-20161008 (A/B) | 20171226-20180101 (A/B) |
| 20170221-20170227 (A/B) | 20161225-20161231 (B/A) | 20161231-20170106 (B/A) | |
| 20170404-20170410 (B/A) | 20181227-20190102 (B/A) | 20170331-20170406 (A/B) | |
| 20170404-20170416 (B/B) | | 20170711-20170717 (B/A) | |
| 20170404-20170422 (B/A) | | 20170927-20171003 (A/B) | |
| 20170703-20170709 (A/B) | | 20171226-20180101 (B/A) | |
| 20171001-20171007 (B/A) | | 20180326-20180401 (A/B) | |
| | | 20180630-20180706 (A/B) | |
| | | 20180928-20181004 (B/A) | |
| | | 20181227-20190102 (A/B) | |

As Fig. 1 shows, the above inputs of DEM and reference velocities are specifically chosen for Greenland Ice Sheet and for
illustration purposes only. For the global processing, including Arctic, Antarctic and all the other areas of the world, e.g. high mountain glaciers, we use the Copernicus DEM GLO-30 (https://spacedata.copernicus.eu/web/cscda/dataset-details?articleId=394198) and ITS_LIVE Version 1 Landsat-derived velocity mosaics (https://its-live.jpl.nasa.gov) as the reference velocities for deriving all our Version 2 ITS_LIVE products. For the Greenland and Antarctic ice sheets, ITS_LIVE Version 1 Landsat-derived velocity mosaics are combined with MEaSUREs Greenland Ice Sheet Velocity Map from InSAR
Data V002 (Joughin et al., 2010; Joughin et al., 2015) and MEaSUREs InSAR-Based Antarctica Ice Velocity Map, Version 2 (Mouginot et al., 2012; Rignot et al., 2017), respectively. All inputs were provided on a common 120 m grid. The NSIDC Sea Ice Polar Stereographic North projection (EPSG code 3413) is used for all areas North of 55° N latitude. For areas South of 56° S latitude the Antarctic Polar Stereographic South (EPSG 3031) projection is used. For the rest of the world we use local Universal Transverse Mercator (UTM) projections. For all map projections, a constant grid posting of 120 m is used to enhance
the product resolution (by 50%) while keeping computational demands within reason. 240 m posting was used for Version 1.0 of the ITS_LIVE image pair products.

### 2.1.2 ITS_LIVE Sentinel-1 Image-Pair Data Product

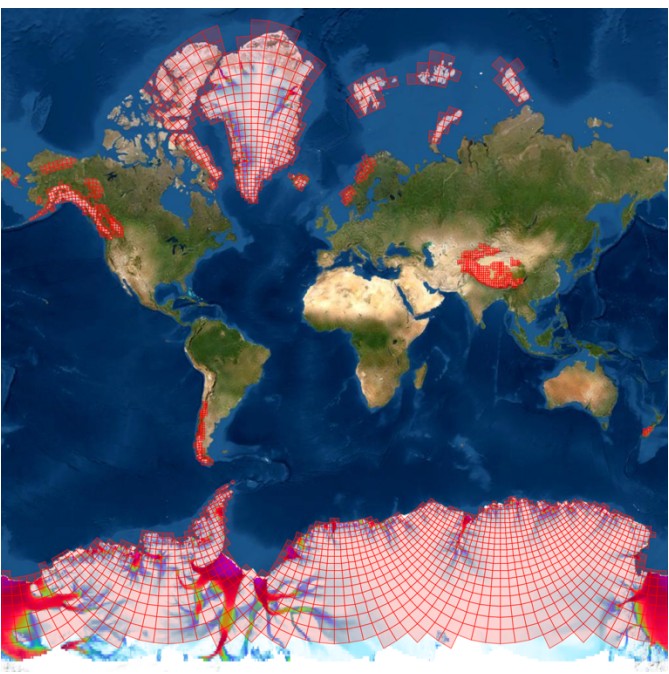

**Figure 2: Global view of the spatial coverage of the ITS_LIVE Version 2 dataset (with Sentinel-1 available between 78° S and 81° N) which include image-pair products, as well as cloud optimized datacubes that contain all ITS_LIVE image-pair products. The ITS_LIVE Version 1.0 Landsat derived ice velocity mosaic map is overlaid onto the global optical image as a reference velocity map, where the red boxes represent the spatial extents of the datacubes.**

Fig. 2 shows the spatial coverage of the ITS_LIVE Version 2 Sentinel-1 image-pair velocity product (https://doi.org/10.5067/0506KQLS6512). The dataset can currently be queried using the ITS_LIVE Data Portal (https://nsidc.org/apps/itslive/) or the by using the ITS_LIVE API tool (https://staging.nsidc.org/apps/itslive-search/docs). In the near future the data will also be accessible through NASA's Earthdata cloud and the National Snow and Ice Data Center. Other access options that facilitate time-series analysis can be found by going to the project website (https://its-live.jpl.nasa.gov).

The products are created using the input dataset as introduced above in Section 2.1.1 and the feature tracking processing chain detailed in Section 2.1.3. Each data granule consists of maps of land ice velocities over the spatial extent of the individual Sentinel-1 image pair at the specific epoch of data acquisition. The image-pair data product provides the most granular data for studying glacier change and is released as an independent data product to the end-users. This product serves as the base for the upcoming ITS_LIVE Version 2 datacube and mosaic products. All of the Sentinel-1A (launched April 3, 2014) and Sentinel-1B (launched April 25, 2016) image pairs with a maximum time separation of 12 days, with a goal of extending to 60 days, are processed into image-pair data products covering the globe (see Fig. 2) within the Sentinel-1 latitude limits (78°

S to 81° N) that includes all large glacierized regions (roughly >25 km$^2$), dependent on data availability and quality. Here, the offset tracking at a given grid point is defined as "valid" if the determined pixel displacement using the current chip passes the NDC filter based on the similarity with adjacent chips, as further discussed in Section 2.2. The spatial extent of the image-pair encompasses the valid offsets for that pair. This means that two granules with the same Sentinel-1 orbit/frame designations may have different extents depending on offset tracking success – if the input images were largely contaminated by temporal decorrelation and/or atmospheric disturbance effects, then the extent of the output can be significantly reduced from the input image extents.

Image pair products span the time period from the launch date of the corresponding sensors (e.g., Sentinel-1A—Sentinel-1A image pairs are available after the Sentinel-1A launch date, Sentinel-1B—Sentinel-1B and Sentinel-1A—Sentinel-1B or Sentinel-1B—Sentinel-1A image pairs are available after the launch date of Sentinel-1B) to present with a project goal of providing < 3 months latency from the date of acquisition. The separation times for the image pairs vary from 6 to 12 days for all sensor combinations, with the goal of extending to 60 days with multiples of 6-day separation dependent on funding. As a rule of thumb, for working with individual image-pair products over fast-flowing glacierized regions where rapid temporal decorrelation of radar signals is expected, using shorter separation times if available (e.g. 6 day; Sentinel-1A—Sentinel-1B or equivalently Sentinel-1B—Sentinel-1A) usually has better data quality than longer ones (e.g. 12 days or larger; Sentinel-1A—Sentinel-1A or Sentinel-1B—Sentinel-1B) due to less temporal decorrelation, which has been investigated by (Friedl et al., 2021; Solgaard et al., 2021). However, for observing slow-moving targets, all image-pair products with various separation times need to be considered to fully capture both the short-term and long-term variability of the velocities, such as in studying the seasonal ice dynamics (Greene et al., 2020).

All output variables are provided on the same 120 m grid in the same map projection as the input dataset detailed in Section 2.1.1. The effective spatial resolution in offset tracking varies spatially with the progressive search chip size (Fig. 1d-e; from 240 m to 1920 m), where fast-flowing glaciers are characterized by smaller chip sizes (lower precision but finer resolution, e.g. 240 m), and the stable or slow-flowing ice uses larger chip sizes (lower resolution but higher precision, e.g. 1920 m).

The output data (Table 2) are packaged in a single NetCDF, using Climate Forecast (CF) Version 1.8 conventions. Individual NetCDF files are between 5 MB and 15 MB in size. The x and y velocities (**vx** and **vy**), velocity magnitude (**v**) and its error (**v_error**), and **chip_size_height**, **chip_size_width**, **interp_mask**, and **img_pair_info** are standard output variables for all ITS_LIVE image-pair velocity products (i.e. both optical and radar). Remaining variables listed in Table 2 are specific to radar data, such as the slant-range (LOS) and azimuth (along-track) velocities (**vr** and **va**) that are provided in native radar viewing geometry. LOS parameters (**M11** and **M12**) are provided in the output for each image pair to facilitate the inversion for x/y horizontal velocity when using two independent slant-range measurements (i.e. one ascending image pair and one descending image pair). This is useful to correct for ionosphere disturbance effects on the azimuth offset (Section 3.2).

Velocities are calculated from imagery that has been map projected. This can introduce scale errors of up to a few percent that are dependent on the projection used and the location of the imagery. Unlike correcting the scale distortion in ITS_LIVE

Version 1 products, for ITS_LIVE Version 2 products (both radar and optical), the scale distortion due to calculating velocities from map projected imagery is not corrected for (i.e. velocities are relative to map coordinates). The implications of this is detailed further in the Version 2 documentation (http://its-live.jpl.nasa.gov/).

### 2.1.3 Processing chain

The Sentinel-1 Single Look Complex (SLC) image pairs are pre-processed using the ISCE software prior to dense offset-

235 tracking, where the two SLC images are precisely co-registered using the satellite orbit geometry. Dense offset-tracking relies heavily on two Python modules: *Geogrid* and *autoRIFT*, followed by a NetCDF packaging program (all of which are open access and available at https://github.com/nasa-jpl/autoRIFT). As described by Lei et al. (2021a), the Sentinel-1 orbit information is used by *Geogrid* to transform all of the input parameters listed in Table 2 from map-projected Cartesian coordinates (Northing/Easting) into radar range/azimuth coordinates (pixel index and displacement). The inputs are translated

to radar coordinates, but not resampled (i.e. remains in original 120 m grid). Inputs are cropped to the spatial extent of the image pair overlap. At this stage, all distances (in m) and velocities (in m yr$^{-1}$) have been converted to pixel distances in range/azimuth, where velocities are multiplied by the time separation between images to convert from a rate to a distance. The approach of defining a regular output grid in map-projected coordinates and then converting to an irregular grid in radar coordinates is somewhat unique, yet powerful. The approach avoids the need for any reprojection of the imagery or resampling

of the derived velocity fields when moving between radar and map coordinates. This eliminates costly interpolation, and resulting artifacts, in the final product.

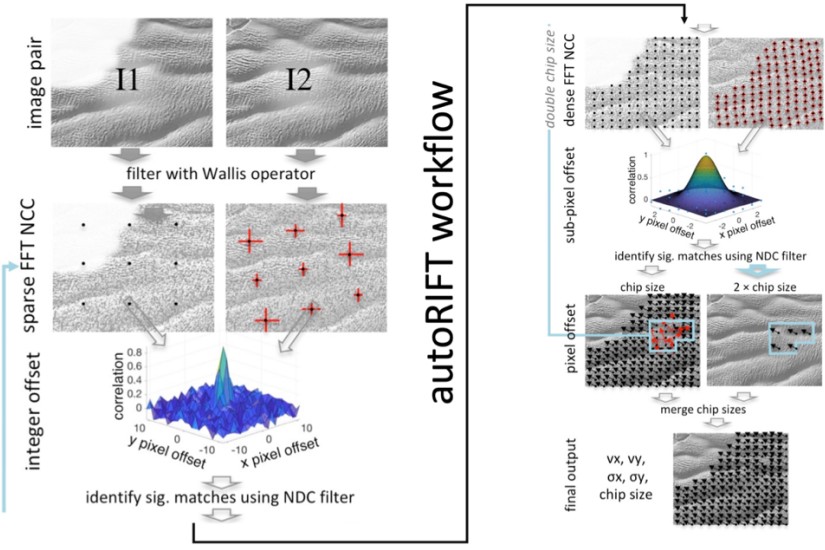

**Figure 3: *autoRIFT* workflow.**

Once inputs (Table 2) have been mapped/converted to radar coordinates/units they are passed to the *autoRIFT* module which preforms the dense range and azimuth offset search between the two images. As illustrated in Fig. 3, *autoRIFT* finds the displacement between two images using a nested grid design, sparse/dense combinative searching strategy, and disparity filtering technique that result in significant performance gains and that are detailed in Gardner et al. (2018) and Lei et al. (2021a). *autoRIFT* then cycles from the minimum to the maximum chip size (Fig. 1d-e) at a multiple of two until a valid offset

match is found, returning the value of the first valid offset (finest resolution). Each progressive chip size is a factor of 2 larger than the previous and is solved on a geographic grid that satisfies a 50% overlap between adjacent search chips of the same size (i.e. a 240 m × 240 m chip is solved on a constant 120 m grid, a 480 m × 480 m chip is solved on a constant 240 m grid). This change in sample rate is termed a "nested grid" design. *autoRIFT* employs the NDC filter to distinguish valid offset matches from random noise. Earlier versions of *autoRIFT* did not permit overlap between adjacent search chips. To

accommodate overlapping chips, adjustments were made to the NDC filter parameters and are detailed in Section 2.2. For each chip size used, a sparse search (1/8 sample rate) is performed to identify and exclude areas that do not pass the NDC filter. A dense search is then performed for all valid areas at the full sample rate, with search center and range informed by the results of the sparse search. This sparse/dense combinative searching strategy substantially improve computational efficiency (i.e. 2 orders of magnitude improvement; as further discussed in Section 2.4). *autoRIFT* further improves efficiency without

sacrificing accuracy or resolution by employing: OpenCV's C++ functions for Normalized Cross-Correlation (NCC), a downstream search routine to reduce the search range, subpixel offset estimation using a Gaussian pyramid upsampling algorithm and, various high-pass filtering options to pre-process the image pair with data compression (Gardner et al. 2018; Lei et al. 2021a). In Section 2.3 we describe the addition of chip size dependent subpixel oversampling ratios that further enhance performance.

Sentinel-1 range and azimuth offset estimates are then calibrated for the subswath and full-swath dependent geolocation biases that are identified for Sentinel-1A/B combinations that are affected by an inter-satellite systematic bias (Section 3.1). Offsets are then converted back to map-projected coordinates/units using *Geogrid*. All of the output data (Table 2) are packed as a NetCDF with accompanying metadata.

**2.2 Fine output grid with improved NDC filter**

One key component of *autoRIFT* that identifies and removes poor feature matches is the implementation of the NDC filter. In this section, we document updates to the NDC filter for handling finer grid spacings with overlapping search chips. As illustrated in Fig. 4a, the original NDC filter assumed chip independence (i.e. no overlap between adjacent chips). However, when oversampling is desired, adjacent chips overlap (e.g. Fig. 4b shows the case of 50% overlap). This results in information

being shared between neighbouring search chips, changing the signal-to-noise statistics. For this reason, we modified the NDC filter parameters to account for the change in the signal-to-noise statistics.

As first developed in Gardner et al., 2018 and also described in Lei et al., 2021a, the NDC filter in *autoRIFT* identifies and removes low-coherence displacement results based on displacement disparity thresholds that are normalized to the local search
distance. This is done to normalize changes in the amplitude of the noise that scales linearly with the search distance. The NDC filter is defined with a few filter parameters: **FracValid** (fraction of valid grid points within the filter window size; default is 8/25, e.g. 8 valid points for a $5 \times 5$ filter window), **FracSearch** (fraction of displacement disparity normalized to search distance; default is 0.2), **FiltWidth** (filter window size; default is 5), **Iter** (number of iterations that the filter is applied; default is 3) and **MadScalar** (multiplicative factor for thresholding the displacement disparity; default is 4). The NDC filter is
applied as a sliding window filter. Here we provide an example of the NDC filter applied using a $5 \times 5$ filter window size:

   a. Normalize the displacement estimates by the search distance;
   b. Compute the normalized displacement disparity in reference to the central grid point for all the grid points within the filter window and count the number of grid points that have displacement disparity smaller than **FracSearch** (threshold of maximum disparity with the default value of 0.2);
c. If the number of grid points in the above step is greater than or equal to **FracValid** $\times$ **FiltWidth**$^2$ (by definition, it is the threshold of valid points within the filter window, e.g., $8/25 \cdot 5^2 = 8$ in this case), the central grid point is retained; otherwise, it is discarded;
   d. A second condition check is then performed: The central grid point is retained if the deviation of its displacement from the median value of this filter window is less than the Median Absolute Deviation (MAD) multiplied by
**MadScalar**. This step is repeated for **Iter** times for the entire displacement field by discarding poor offset matches in each iteration.
   e. If all the above conditional checks are passed, the central grid point is retained; otherwise, it is discarded.

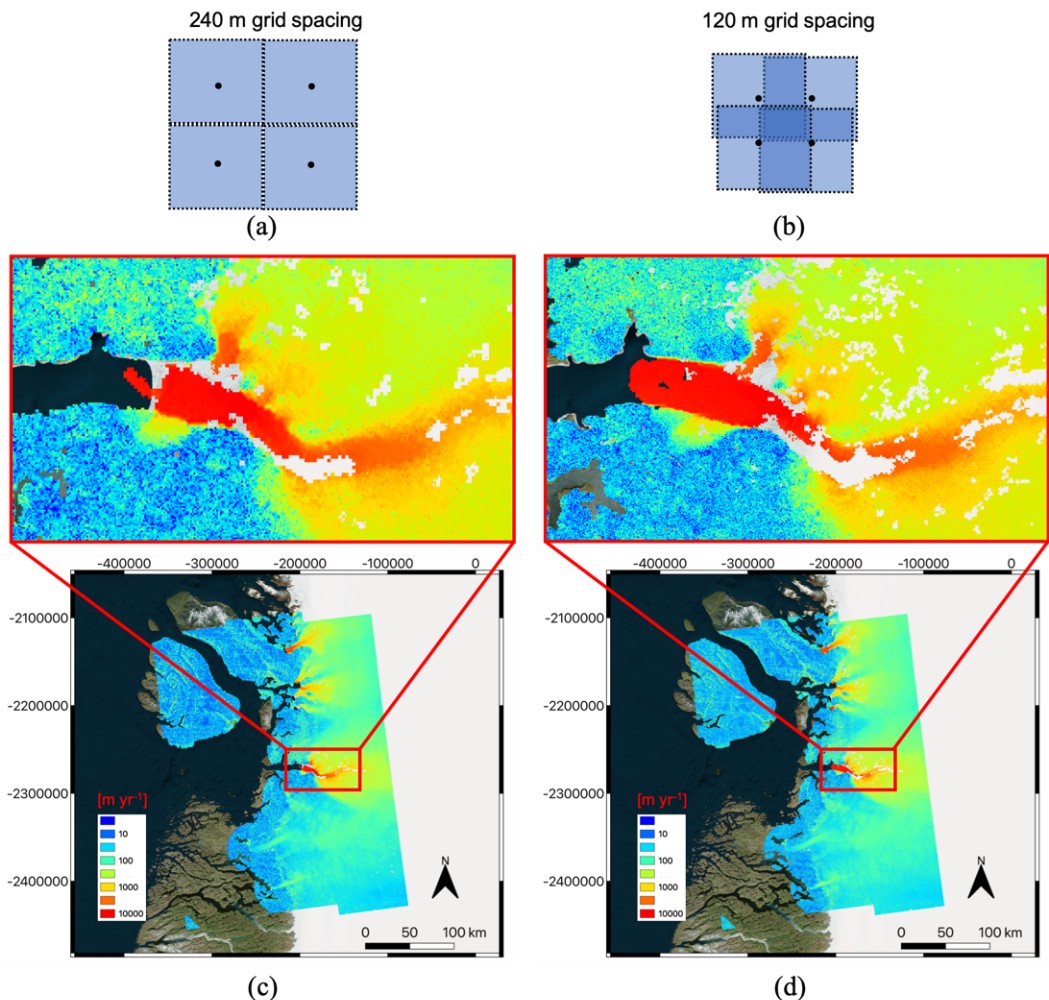

**Figure 4: Comparison between a 240 m posting grid and a 120 m posting grid: (a) 240 m grid with non-overlapping search chips, (b) 120 m grid with 50% overlapping search chips, (c)** *autoRIFT***-estimated ice velocity magnitude for 240 m grid, (d)** *autoRIFT-***estimated ice velocity magnitude for 120 m grid. The same minimum chip size of 240 m is used for both cases. For (c) and (d), the Sentinel-1 image pair 20170404-20170410 is used in Region 1 (with the closeup at the Jakobshavn Isbræ Glacier).**

The filter parameters need to be adjusted when there is overlap between adjacent chips to maintain the same filter performance as in the non-overlap scenario. For the 50% overlap case illustrated in Fig. 4b, the default **FracSearch** value of 0.2 can be used but **FracValid** needs to be made larger due to the inter-dependence of neighbouring offsets. As an extreme case, with 0% overlap, **FracValid** should be the same as the default value for the non-overlap case; with 100% overlap, **FracValid** should be 1. Therefore, considering these extreme cases, and through trial and error, we found that the following formula works well for all cases considered:

$$FracValid_{\text{w\_overlap}} = FracValid_{\text{wo\_overlap}} \times (1 - overlap) + overlap^2, \tag{1}$$

where $FracValid_{\text{w\_overlap}}$ and $FracValid_{\text{wo\_overlap}}$ are the **FracValid** parameters with and without overlap, and $overlap$ is the percentage of the overlap (e.g. 50% in Fig. 4b). Since the adjacent grid points share information, the filter window size

needs to be enlarged by a factor of $1/overlap$ in order to have an equivalent number of independent samples within a filter window:

$$FiltWidth_{w\_overlap} = round\left[\left(FiltWidth_{wo\_overlap} - 1\right)/overlap + 1\right],$$  (2)

where $FiltWidth_{w\_overlap}$ and $FiltWidth_{wo\_overlap}$ are the **FiltWidth** parameters with and without overlap, and round$[\cdot]$ is the round to integer operation. Using Eq. (1) and (2), the NDC filter performance for the overlapping chips is comparable to the non-overlap scenario. Fig. 4c demonstrates the ice velocity magnitude from the Sentinel-1 image pair 20170404-20170410 centered over the Jakobshavn Isbræ Glacier in Region 1 (Fig. 1a) using the earlier 240 m posting grid with non-overlapping

search chip of size 240 m (Fig. 4a). Fig. 4d shows results for the 120 m posting grid using a search chip of size 240 m with 50% overlap between adjacent search chips (Fig. 4b). From the results in Fig. 4c-d, it can be seen that the region of interest (ROI) representing valid pixels between the two cases are very similar, implying the comparable filter performance, while the 120 m grid indeed provide higher resolution estimates without sacrificing signal-to-noise over the ROI.

## 2.3 Search-chip-size dependent subpixel oversampling ratio

*autoRIFT* identifies the integer offset between two images as the maximum NCC value for all possible locations of the search chip (subset of image 2) within the search window (subset of image 1). To identify the subpixel component of the offset, an oversampled surface is fit to the pixel NCC values to create a smooth surface from which the subpixel offset can be estimated. Careful consideration must be taken on how the surface is fit, as some approaches can lead to biases in offset estimates (Stein et al., 2006). To minimize such bias, *autoRIFT* employs a Gaussian pyramid upsampling algorithm. Surface fitting is

computationally expensive, increasing substantially with the oversampling ratio. Therefore, subpixel peak finding should consider the inherent achievable precision of the underlying data in determining the optimal oversampling ratio. An oversampling ratio that is too coarse will result in less precise data, while an oversampling ratio that is too fine will result in unnecessary computational overhead. In addition, there is an inherent relationship between the size of the search chip and the maximum achievable precision; larger chip size results in higher achievable precision and vice versa. Therefore, it is desirable

to select an optimal trade-off between efficiency and the precision of the subpixel oversampling ratio as a function of chip size, as in Lei et al. (2021a).

We consider four chip sizes: 240 m, 480 m, 960 m and 1920 m, and four subpixel oversampling ratios: 1/16, 1/32, 1/64 and 1/128. To investigate the relation between chip size dependent accuracy and oversampling ratio we select the Sentinel-1 image

pair 20171226-20180101 in Region 2. Region 2 represents slow-moving ice surface with no detectable gradient in flow and all four chip sizes result in valid offset tracking results. Results determined using an oversampling ratio of 1/128 are considered most precise. Fig. 5 shows the x-velocity (**vx**) for various chip sizes using an oversampling ratio of 1/128. Note that the subswath bias (visible at subswath boundaries) with azimuth streaks that are noticeable in Fig. 5 are due to the Sentinel-1 systematic geolocation bias and ionosphere delay effects, both of which are discussed and accounted for using the methods

presented in Section 3. From Fig. 5 it can be seen that smaller chip sizes (240 m and 480 m) have a high standard deviation (noise), while larger chip sizes (960 m and 1920 m) have lower standard deviation (more precise) in the displacement estimates.

The question we now try to answer is: Is the computational cost of a finer oversampling ratio justified by an improvement in precision? To determine this balance, we examine changes in the standard deviation as a function of oversampling ratio for each chip size. A summary of the results is provided in Table 4.

**Table 4.** Selection of subpixel oversampling ratio for various chip sizes. The first four rows show the results for the highest oversampling ratio (1/128) which is used to characterize the maximum achievable precision. The last four rows show the results for the lowest oversampling ratio that provides negligible degradation (1-3%) in precision. The column of "nearest oversampling ratio" is selected to match the maximum achievable precision and thus determine the optimal oversampling ratios to be used for each chip size (first column of the last four rows).

| Oversampling Ratio | Chip Size [m] | Runtime [sec] | x/y precision [pixel] | Nearest oversampling ratio |
|---|---|---|---|---|
| 1/128 | 1920 | 52.9 | 0.0086/0.0097 | 1/128≈0.0078 |
| 1/128 | 960 | 83.5 | 0.0118/0.0122 | 1/128≈0.0078 or 1/64≈0.0156 |
| 1/128 | 480 | 200.8 | 0.0155/0.0185 | 1/64≈0.0156 |
| 1/128 | 240 | 661.5 | 0.0321/0.0392 | 1/32=0.0312 |
| 1/128 | 1920 | 52.9 | 0.0086/0.0097 | — |
| 1/128 | 960 | 83.5 | 0.0118/0.0122 | — |
| 1/64 | 480 | 110.4 | 0.0158/0.0188 | — |
| 1/32 | 240 | 188.1 | 0.0332/0.0401 | — |

From our analysis we find that a coarser oversampling ratio of 1/32 (1/64), for a chip size of 240 m (480 m), results in a modest increase in noise (<3%) with up to a factor of 3 improvement in performance (Table 4). We therefore select these oversampling ratios for the smaller chip sizes. For a chip size of 960 m and 1920 m we see a substantial (30% for 960 m and 80% for 1920 m) increase in noise with a decrease in oversample ratio (1/64). Therefore, we select an oversample ratio of 1/128 for these larger chip sizes. This intelligent oversampling ratio is adopted in *autoRIFT* for generating the ITS_LIVE Sentinel-1 offset tracking products. The optimal oversampling ratio is dependent on the information content of the input imagery and would require a similar analysis, as conducted here, for application to other satellite missions. For Landsat-8 data *autoRIFT* uses an oversampling ratio of 1/16, 1/32, 1/64 and 1/64 for chip sizes of 240 m, 480 m, 960 m and 1920 m, respectively. A coarser oversampling ratio leads to coarser velocity discretization. This can be seen when plotting chip size dependent histograms of **va** and **vr** in which values are clustered according to the oversampling ratio. A smoother histogram can be achieved with a finer oversampling ratio but, due to the limitations of the data, a finer oversampling ratio will not achieve higher accuracy. The precision and accuracy can be increased through postprocessing of the data (e.g. spatial or temporal smoothing).

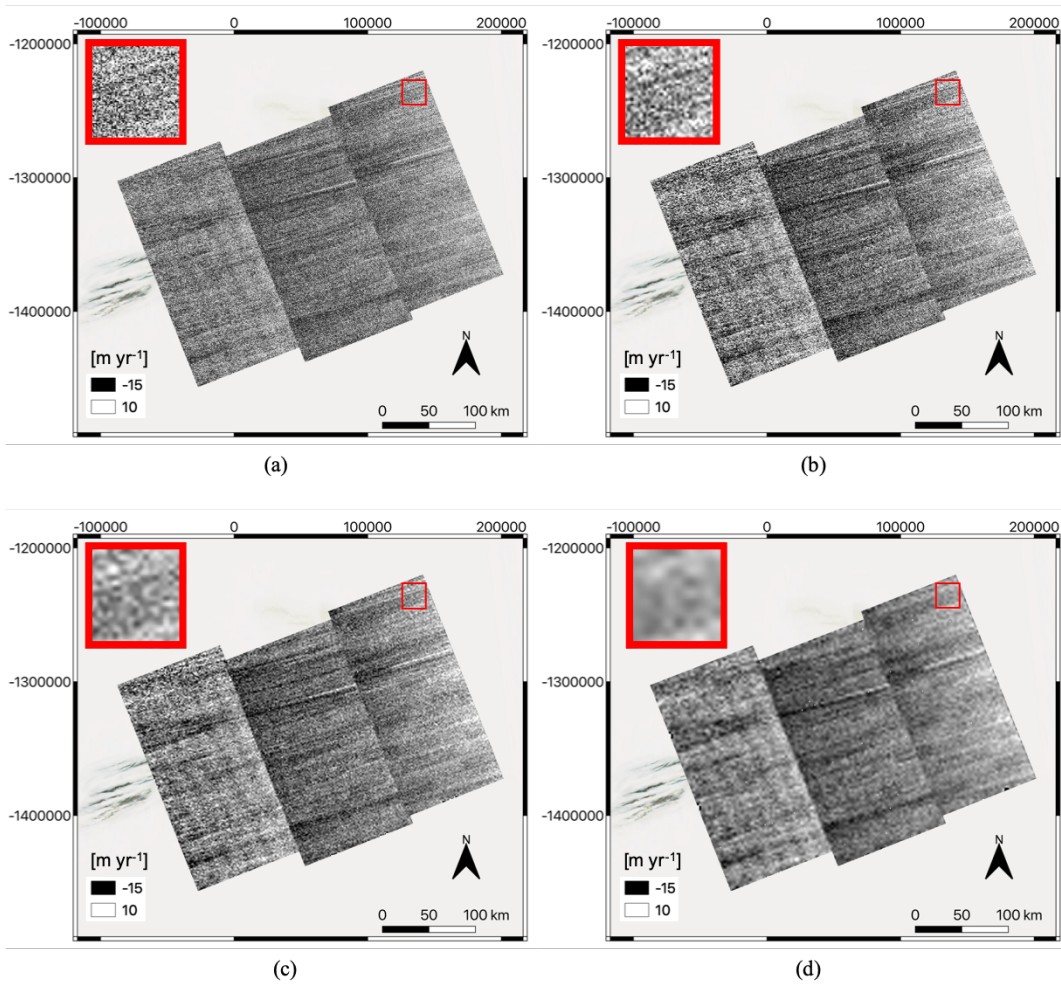

**Figure 5: x-velocity from offset tracking using a chip size of: (a) 240 m, (b) 480 m, (c) 960 m, and (d) 1920 m. The same subpixel oversampling ratio of 1/128 is used for all cases. The Region 2 Sentinel-1 image pair 20171226-20180101 is used.**

### 2.4 *autoRIFT* runtime analysis

As previously shown by Lei et al. (2021a), *autoRIFT* outperforms the widely-used "dense ampcor" feature tracking algorithm (CPU version) that is the default feature tracker included in NASA/JPL's ISCE software, with two orders of magnitude improvement in efficiency and >20% improvement in accuracy. Here we expand the apples-to-apples comparison between *autoRIFT* and ampcor that was conducted by Lei et al. (2021a). We use 7 Region-1 Sentinel-1 image pairs with the same *autoRIFT* and ampcor setting as used by Lei et al. (2021a: Table 4). The runtime and accuracy improvements of *autoRIFT* compared to ampcor are illustrated in Fig. 6 as a function of the % valid ROI (i.e. the % of valid pixels returned by *autoRIFT*). In Fig. 6a, an exponential function was fit to the runtime data points with respect to ROI. The figure shows that *autoRIFT* is about 150 times faster than ampcor for Sentinel-1 image pairs with high correlation between images (large ROI) and up to 208 times faster when there is low correlation between images (low ROI). This increase in runtime improvement with a decrease

in ROI is due to *autoRIFT*'s sparse search that excludes areas of low correlation before executing a dense search. Regarding the accuracy of both feature trackers, we refer to the standard deviation of x/y pixel displacements over stable surfaces (Fig.

1f). Here, x/y represents the dimensions in the native radar image coordinates, i.e. range/azimuth. Since dense ampcor does not employ any filtering to remove bad matches, we calculated the error of dense ampcor results wherever *autoRIFT* produced reliable estimates. As shown in Fig. 6b, *autoRIFT* provides an improvement in accuracy on the order of 20-50% (33% when averaged across all 7 image pairs). Some of the improvement can be attributed to *autoRIFT*'s ability to narrow the search range of the dense search based on information gained from the sparse search (a form of regularization). Different approaches to

locating the subpixel displacement may also contribute to *autoRIFT*'s improved accuracy (e.g. Gaussian pyramid upsampling algorithm).

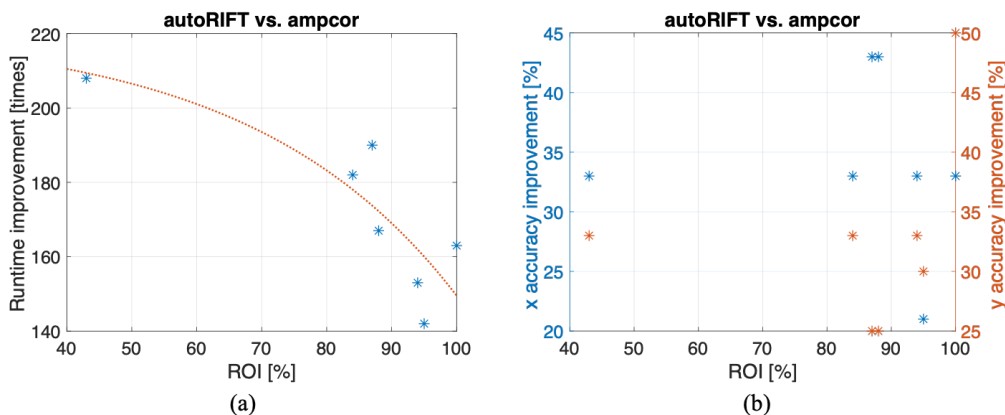

(a)        (b)

**Figure 6: Apples-to-apples comparison between *autoRIFT* and ampcor: (a) runtime improvement, (b) x/y (range/azimuth) accuracy**

**improvement. Both *autoRIFT* and ampcor use the same setting for offset tracking as in Lei et al. (2021a): chip size of 64 × 64, grid spacing of 64 × 64, and search limit of 62 × 16. The 7 Sentinel-1 image pairs in Region 1 (Frame 222 only) are used.**

## 3 ITS_LIVE Sentinel-1 image pair product calibration and error correction

In this section we demonstrate approaches to calibrate and correct for errors in the ITS_LIVE Sentinel-1 image pair product. Section 3.1 discusses the systematic geolocation bias calibration (both subswath and full-swath dependent). In Section 3.2, we

introduce a method of correcting for the ionosphere disturbance effects that contaminate radar azimuth measurements, which uses the radar LOS measurements from two image pairs with differing acquisition geometries (i.e. ascending and descending).

### 3.1 Geolocation bias calibration

As pointed in Gisinger et al. (2021), Sentinel-1's azimuth Doppler Frequency Modulation (FM) rate can be offset compared with the modelled value for each of the three subswaths in the TOPS acquisition mode because of the use of constant terrain

height for FM rate computation for extended areas of observed terrain (spatial extent of the SAR image). In reality, each of the three subswaths has its own azimuth FM rate. This systematic mismatch can cause both full-swath and subswath dependent

pixel shift (geolocation bias) along the radar azimuth direction when focusing radar signals. Full-swath and subswath dependent systematic geolocation errors along the radar range direction also result from the bistatic nature of the antenna, i.e., transmitting and receiving are not simultaneous, which is neglected in the current SLC processing and thus introduces a residual Doppler shift error when focusing the radar pulses in the range direction (Gisinger et al., 2021). Several studies (Gisinger et al., 2021; Solgaard et al., 2021; Schubert et al., 2017) have investigated the full-swath dependent range/azimuth geolocation bias of Sentinel-1 IW products, and also mentioned the existence of the subswath dependent geolocation bias. At the time of writing, besides the subswath dependent range bias estimates (Zhang et al., 2022), we are not aware of any existing subswath dependent geolocation bias (both range and azimuth) correction. In the following subsections we demonstrate the calibration of both types of geolocation bias for the ITS_LIVE Sentinel-1 ice velocity products.

### 3.1.1 Subswath dependent geolocation bias

When working with Sentinel-1 image pairs, systematic geolocation biases cancel out in the resulting offset maps when using data from the same sensor (both from Sentinel-1A or both from Sentinel-1B); however, this is not the case when using a combination of the two (one from Sentinel-1A and the other from Sentinel-1B). For this reason, we focus on corrections that need to be applied when image pairs are composed of images from differing sensors.

As mentioned in Section 2.3 (Fig. 5), there is subswath dependent velocity bias between the three subswaths. We illustrate this effect by showing both the raw slant-range and azimuth velocity products ($\mathbf{vr}$ and $\mathbf{va}$ from Table 2) in Fig. 7a-b for the Sentinel-1 pair 20171226-20180101 in Region 2, where the range-dependent variation of the velocity (by averaging each range line/bin) is found in Fig. 7c. Fig. 7c clearly demonstrates the subswath mismatch. To correct for the subswath mismatch, we select two areas in the interior of Greenland (Region 2 and 3) with 14 Sentinel-1A/B image pairs (in Table 3) that contain small and smooth gradients in ice motion and ionospheric effects, and are thus more suitable for calibration of the biases. Only 11 out of the 14 pairs are eventually shown in Fig. 8 after eliminating 3 noisy pairs contaminated by strong ionosphere scintillation (Jiao et al., 2013). We calculate the inter-subswath bias by differencing the median values on either side of the subswath boundaries of the slant-range and azimuth offset. The results of this analysis are shown in Fig. 8.

From Fig. 8, the average offset bias between subswath 2 and 1 are -0.010 pixel (slant-range) and 0.019 pixel (azimuth), while the estimates for subswath 3 and 2 are -0.007 pixel (slant-range) and 0.006 pixel (azimuth), which uses the pair convention of Sentinel-1A being acquired prior to Sentinel-1B. These inter-subswath bias estimates are similar to the numbers reported in the other works of the literature over various regions of the globe, e.g., -0.0082 pixel (slant-range) between subswath 2 and 1, and -0.0073 pixel (slant-range) between subswath 3 and 2 (Zhang et al., 2022). Therefore, they are considered systematic biases and thus used in generating our ITS_LIVE Sentinel-1 image pair products for calibrating the subswath dependent geolocation bias. The performance of the correction is illustrated in Fig. 7d-f as well, where both the velocity maps and the range-dependent variation curve show greatly improved matching between subswaths (i.e. there is little to no step change in velocity when transitioning between subswath boundaries).

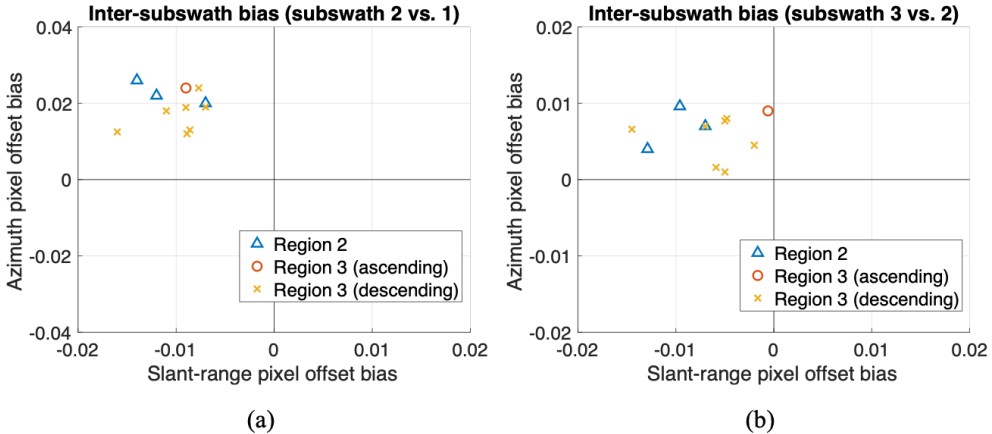

**Figure 7: Illustration of subswath dependent geolocation bias and calibration: (a) raw slant-range velocity, (b) raw azimuth velocity, (c) range-dependent variation of raw velocity (averaged for each range line/bin), (d) corrected slant-range velocity, (e) corrected azimuth velocity, (f) range-dependent variation of corrected velocity (averaged for each range line/bin). The Sentinel-1 image pair 20171226-20180101 in Region 2 is used here.**

**Figure 8: Inter-subswath bias of slant-range and azimuth offsets: (a) subswath 2 versus 1, (b) subswath 3 versus 2. The 11 Sentinel-1A/B image pairs in Region 2 and Region 3 are shown here.**

### 3.1.2 Full-swath dependent geolocation bias

After the subswath dependent geolocation error correction is applied, we apply additional corrections to remove full-swath dependent biases. As reported in previous studies (Gisinger et al., 2021; Solgaard et al., 2021; Schubert et al., 2017), the geolocation error for the full swath can be measured by comparing the SLC image pixel geolocations or resulting offset-tracking velocity to ground control points, e.g., array of corner reflectors or stable terrain. Locally determined estimates can then be applied to all the Sentinel-1 image pairs as a static calibration assuming that the error is systematic in nature. Conceptually, this calibration is similar to referencing an interferogram to a known reference point (e.g. GPS station) in InSAR analysis. In this paper, we adopt the similar approach but apply a unique correction to each Sentinel-1 image pair, by separately calibrating to any overlapping stable surfaces, as has been done in our previous work (Gardner et al., 2018; Lei et al., 2021a). We determine the calibration surface using a "stable surface" mask (Fig. 1f) that is defined as any area consisting of ice-free terrain and/or slow-moving ice defined using an a-priori reference velocity ($< 15$ m yr$^{-1}$: Fig. 1f). When image-pair offsets do no intersect any valid "stable surface" we calibrate the slowest 25% velocity magnitude, as defined by a-priori velocity field, to an a-priori reference velocity. We examine the performance of these two approaches for full-swath dependent geolocation bias calibration using all 20 Sentinel-1A/B image pairs (Table 3) in Region 1, 2 and 3. Results are illustrated in Fig. 9.

The mean "stable surface" mask calibrated velocity bias is -4.1 m yr$^{-1}$ in slant range and -26.8 m yr$^{-1}$ in azimuth using the pair convention of Sentinel-1A being acquired prior to Sentinel-1B with a 6-day time separation. Note that in this paper, we report the velocity calibration bias (rather than pixel offset bias in Section 3.1.1 or offset bias in meters) for the full-swath dependent geolocation bias, in order to better compare with the values reported in the literature as shown below. One can convert from one to another by using the 6-day time separation and/or slant-range/azimuth pixel size. Using the slowest 25% velocities, the mean velocity bias estimated to be -4.0 m yr$^{-1}$ in slant range and -27.1 m yr$^{-1}$ in azimuth. Hence, by using a stable surface mask (which is available in most regions) or the slowest 25% velocities, our dynamically calibrated estimates of geolocation bias are comparable to those reported in previous work over various regions of the globe and thus confirmed to be systematic biases, e.g. -8.8 m yr$^{-1}$ in slant range and -28.8 m yr$^{-1}$ in azimuth (Solgaard et al., 2021), -9.7 m yr$^{-1}$ in slant range and -24.4 m yr$^{-1}$ in azimuth for the Extended Timing Annotation Dataset (ETAD) correction (Gisinger et al., 2021), as well as the -5.3 m yr$^{-1}$, -6.4 m yr$^{-1}$, -7.5 m yr$^{-1}$ in slant range for each of the three subswaths (Zhang et al., 2022). Therefore, these bias estimates are used in generating our ITS_LIVE Sentinel-1 image pair products for calibrating the full-swath dependent geolocation bias.

The small difference between our estimates and those reported in literature can be due to the fact that we applied the subswath dependent geolocation bias correction first. We note that the method of using the slowest 25% provides good estimates of the velocity bias, almost equivalent to those by using an external stable surface mask. In the processing, we only calibrate to the slowest 25% when there is insufficient overlap with the "stable surface" mask. Note the above velocity bias estimates are specifically referenced to 6-day Sentinel-1A/B image pairs with Sentinel-1A acquired prior to Sentinel-1B.

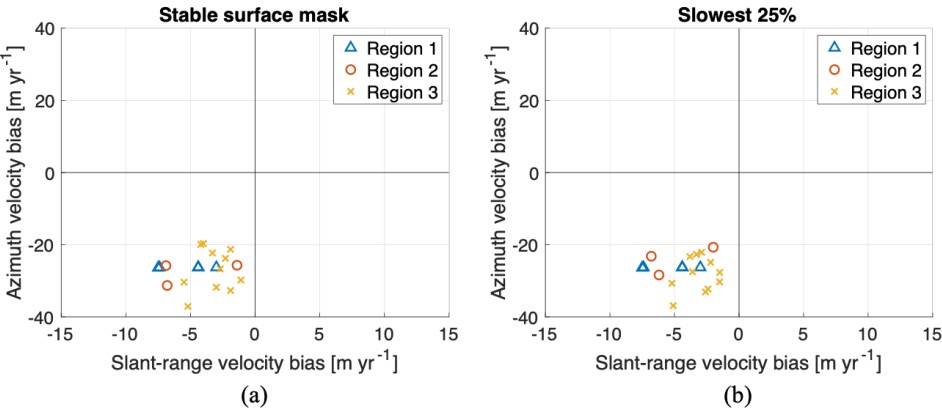

(a)                                              (b)

**Figure 9: Full-swath geolocation bias-induced slant-range and azimuth velocity bias: (a) with stable surface mask, (b) considering the slowest 25% as stable surfaces. The 20 Sentinel-1A/B image pairs in Region 1, 2 and 3 are used. We use the pair convention of Sentinel-1A acquired prior to Sentinel-1B with 6-day time separation.**

Scalar full-swath dependent bias corrections are applied and stored as **stable_shift_mask** and **stable_shift_slow** for each of the velocity variable (e.g., **vx**, **vy**, **vr** and **va**) in our ITS_LIVE Sentinel-1 product (Table 2). After the geolocation corrections are applied, the uncertainty in the velocities are calculated (Lei et al., 2021a) over the intersecting "stable surfaces" area when there is sufficient overlap, otherwise the calculations are made for the slowest 25%. Uncertainties are taken as the standard error between calculated and the reference velocities, as described in Gardner et al. 2018 and elaborated in Lei et al., 2021a. Hence, these estimated uncertainties are assigned as scalar attributes (named "error") to each of the 2-D velocity fields (**vx**, **vy**, **vr** and **va**). However, the uncertainty of the velocity magnitude (**v**), namely **v_error**, must be estimated using the following error propagation formula:

$$\mathbf{v\_error} = \sqrt{\left(\frac{\mathbf{vx}}{\mathbf{v}} \cdot \mathbf{vx\_error}\right)^2 + \left(\frac{\mathbf{vy}}{\mathbf{v}} \cdot \mathbf{vy\_error}\right)^2}, \tag{3}$$

where "vx_error" and "vy_error" are scalar uncertainties of each 2-D velocity field (**vx** and **vy**) estimated over stable (or slowest 25%) surfaces, and all of **vx**, **vy** and **v** are 2-D velocity fields, implying that the velocity magnitude uncertainty **v_error** is a 2-D field as well. In Section 4, we will show the uncertainty estimates for the 2-D velocity field of **vx** and **vy**, along with the uncertainty of the velocity magnitude (**v**) calculated using Eq. (3), at different test sites of the globe using a large amount of ITS_LIVE Version 2 Sentinel-1 image-pair products.

### 3.2 Ionosphere correction: ascending/descending combined velocity

Remaining velocity errors are dominated by atmosphere delay effects, particularly in the polar regions (Nagler et al., 2015; Solgaard et al., 2021). A common ionosphere effect on the offset tracking velocity products is azimuth pixel shifts due to the linear along-track variation of the ionosphere phase delay within the synthetic aperture in SAR processing. The shift is proportional to the linear rate of the ionosphere phase delay and inversely to the azimuth FM rate of the SAR platform (Meyer et al., 2006; Liang et al., 2019). This azimuth pixel shift usually results in long stripe-like artifacts, also called "azimuth streaks", in SAR-derived offset tracking maps. Such errors have been widely reported in the literature (Joughin et al., 1998;

Gray et al., 2000; Joughin, 2002; Strozzi et al., 2008; Joughin et al., 2010; Mouginot et al., 2012; Rignot and Mouginot, 2012; Sánchez-Gámez and Navarro, 2017; Joughin et al., 2017; Liao et al., 2018). Traditional methods for removing these azimuth streaks are: stacking or weighted averaging of multiple velocity time-series estimates (Rignot and Mouginot, 2012; Joughin et al., 2017), and combining InSAR LOS phase measurements from ascending and descending passes (Joughin et al., 1998; Sánchez-Gámez and Navarro, 2017).

Note the uncertainty of offset tracking velocity is roughly 5 times worse for azimuth velocities than for the range ones when using TOPS mode Sentinel-1 image pairs, because the azimuth resolution is roughly 5 times coarser than in range. However, as mentioned above, the impact of ionosphere phase delay on offset tracking velocity that is more severe for azimuth velocities is irrespective of the resolution, but because of the linear along-track variation of the ionosphere phase delay within the synthetic aperture of the SAR processing. Here we examine an approach to remove azimuth streaks in the ITS_LIVE Sentinel-1 image pair products. This approach exploits the LOS (slant range) measurements from the SAR acquisitions, which has minimal impact from ionosphere disturbance.

With the LOS measurement from a single Sentinel-1 image pair (i.e. 1-D observation of displacement), the 2-D flow field is indeterminant. In the case of two image pairs with differing acquisition geometries (i.e. ascending and descending), we are provided with two independent LOS measurements from which the 2-D flow field can be determined. This approach has been applied to InSAR LOS phase as well as range offset measurements (Joughin et al., 1998; Joughin et al., 2018; Sánchez-Gámez and Navarro, 2017). Here we demonstrate how to apply this correction approach to the offset-tracking velocity layers in our ITS_LIVE Sentinel-1 image pair products.

As described in Section 2.1.3 and in Lei et al. (2021a), the *Geogrid* module provides a look-up table of 2×2 conversion matrix between range/azimuth pixel displacements and x-/y-direction velocity fields. Here, the slope parallel assumption (Joughin et al., 1998) is adopted, where surface displacement is assumed to be parallel to the local surface. Therefore, the z-direction (vertical) velocity can be further expressed as a function of the x-/y-direction velocities. Note this assumption becomes problematic over regions of known non-trivial vertical flow, e.g., slope and/or elevation changes.

Below we define the two elements from the first row of the conversion matrix as **M11** and **M12** (see Table 2; denoted as $M_{11}$ and $M_{12}$). The conversion matrix elements, **M11** and **M12**, are stored in the final NetCDF file of the ITS_LIVE Sentinel-1 image pair product with a 32-bit floating point to 16-bit integer data compression. The LOS (slant-range) measurement of pixel displacement (denoted by $D_r$) can be related to the x-/y-directional velocity (**vx** and **vy**; denoted by $v_x$ and $v_y$) via the following relationship:

$$D_r^a = M_{11}^a v_x + M_{12}^a v_y, \tag{4}$$

for the ascending pair (with the superscript "a") and

$$D_r^d = M_{11}^d v_x + M_{12}^d v_y, \tag{5}$$

for the descending pair (with the superscript "d"). Note Eq. (4) and Eq. (5) formulate a system of two linear equations with two unknowns. By inverting these two equations, we can solve for the velocity fields in map coordinates:

$$\begin{pmatrix} v_x \\ v_y \end{pmatrix} = \frac{1}{(M_{11}^a M_{12}^d - M_{12}^a M_{11}^d)} \begin{bmatrix} M_{12}^d & -M_{12}^a \\ -M_{11}^d & M_{11}^a \end{bmatrix} \begin{pmatrix} D_r^a \\ D_r^d \end{pmatrix}. \tag{6}$$

Thus, given two ITS_LIVE Sentinel-1 image pair products (i.e. NetCDF files), one is provided with all of the parameters needed to use Eq. (6) to calculate the map projected velocities solely from LOS measurements. Fig. 10 provides an example of this approach applied to ascending and descending Sentinel-1 image pairs both acquired on 20171226-20180101 in Region 3 (Table 3).

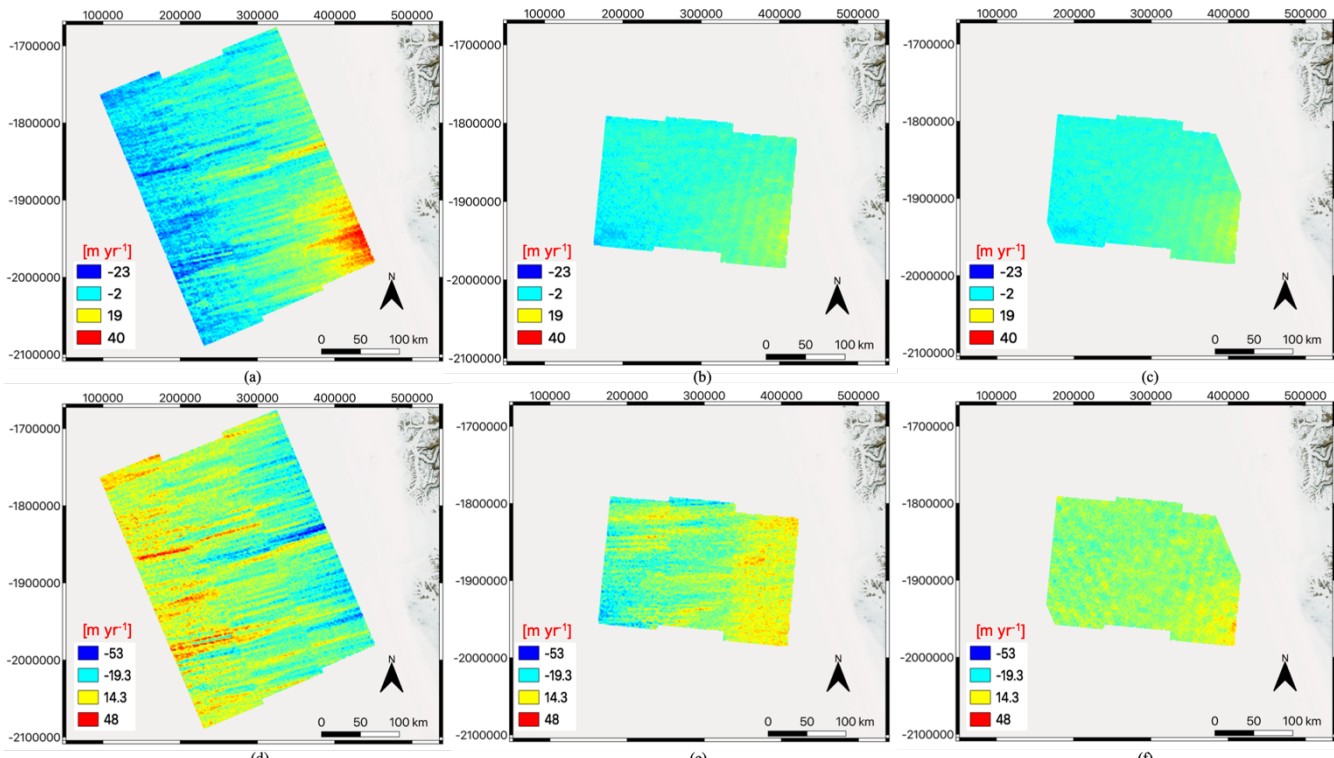

**Figure 10: Ionosphere correction for ascending and descending acquisition geometry: (a) ascending x velocity, (b) descending x velocity, (c) ascending/descending combined x velocity, (d) ascending y velocity, (e) descending y velocity, (f) ascending/descending combined y velocity. The ascending/descending Sentinel-1 image pairs were both acquired on 20171226-20180101 in Region 3.**

From Fig. 10a and Fig. 10d, it can be seen that the x and y velocities from the ascending pair are severely contaminated by the ionosphere disturbance. Streaking is apparent in both x and y component velocities because the inclination of the ascending orbit projects some amount of the azimuth streaking into both coordinate directions. Fig. 10b and Fig. 10e demonstrate the velocities for the descending pair. Since the descending orbit has a small inclination, the ionosphere disturbance is only prominent in the y-direction velocity. Fig. 10c and Fig. 10f show the final velocity fields obtained by combining the overlaps

from both ascending and descending LOS image pair measurements using Eq. (6). Using only LOS displacements significantly reduces the azimuth streaks. The uncertainty reduces from 8.6 m yr$^{-1}$ (ascending) or 3.7 m yr$^{-1}$ (descending) to 3.4 m yr$^{-1}$ (by 9-61%) for the x-direction velocity and from 12.9 m yr$^{-1}$ (ascending) or 10.9 (descending) to 5.6 m yr$^{-1}$ (by 49-57%) for the y-direction velocity. Therefore, significant reductions in velocity error can be achieved, when provided with two ITS_LIVE Sentinel-1 image pair products (i.e., NetCDF files only) acquired from ascending and descending geometry.

## 4 Validation

We validate the global ITS_LIVE Sentinel-1 image pair products, against other publicly available products, for three typical locations: Jakobshavn Isbræ Glacier in Greenland (Section 4.1), Pine Island Glacier in the Antarctic (Section 4.2), and Malaspina Glacier in Alaska (Section 4.3).

### 4.1 Jakobshavn Isbræ Glacier

We first validate the ITS_LIVE Sentinel-1 image pair products over the fast-flowing Jakobshavn Isbræ Glacier (N 69°, W 50°; Fig. 11), located in south-west Greenland. We compare the ITS_LIVE Sentinel-1 velocities to ITS_LIVE Version 2 image pair products acquired from optical sensors (Landsat-8 and Sentinel-2) as well as the PROMICE ice velocities (Solgaard and Kusk, 2021), which are also produced from Sentinel-1 data using the offset-tracking method (Solgaard et al., 2021). Jakobshavn Isbræ Glacier is the fastest glacier draining the ice sheet. The glacier experienced successive break-up events during the late 1990s, and started to speed up afterwards, with the maximum velocity around 15 km/year and an overall seasonal variability of 8 km/year (Lemos et al., 2018). Jakobshavn Isbræ is the fasted moving glacier on Earth and thus serves as a challenging test case for validating the ITS_LIVE Sentinel-1 image-pair products.

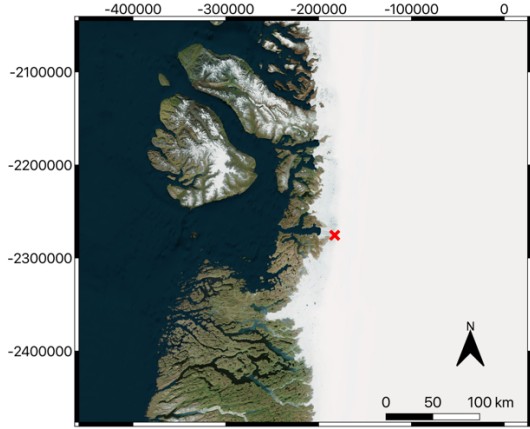

**Figure 11: Optical image of the test site at the Jakobshavn Isbræ Glacier in Greenland, where the red cross marks the location over the fast-moving glacier outlet for the validation of the ITS_LIVE Sentinel-1 image pair product.**

As a qualitative comparison, we show the ITS_LIVE Sentinel-1 image pair product (acquired on 20170404-20170410) along with the PROMICE product in Fig. 12. Since the PROMICE product has a temporal resolution of 24 days, to maximize the

temporal overlap, we used the PROMICE product that is temporally averaged between 20170323 and 20170416. The
difference in grid spacing and effective spatial resolution is apparent between our ITS_LIVE Version 2 product (grid spacing
of 120 m; effective spatial resolution of 240 m to 1920 m) and the PROMICE product (grid spacing of 500 m; effective spatial
resolution of 800 m to 900 m). It can also be seen from Fig. 12 that the PROMICE product has been smoothed/filtered spatially
as well as averaged temporally by mosaicking all the 6-day and 12-day image pair product within the 24-day temporal
resolution window. This is exaggerated when looking at the site overview subfigure of Fig. 12b, where the northern part of the
image shows strong azimuth streaks and the southern part has a discontinuity over stable surfaces. Due to this temporal
mosaicking, there is a difference between the two products for the slowest moving velocities over stable surfaces, i.e., ~20
m/yr. The fast-moving ice velocities (e.g. over the glacier outlet in the closeup of Fig. 12) look very similar between the two
products, with improved coverage provided by the PROMICE temporal averaging.

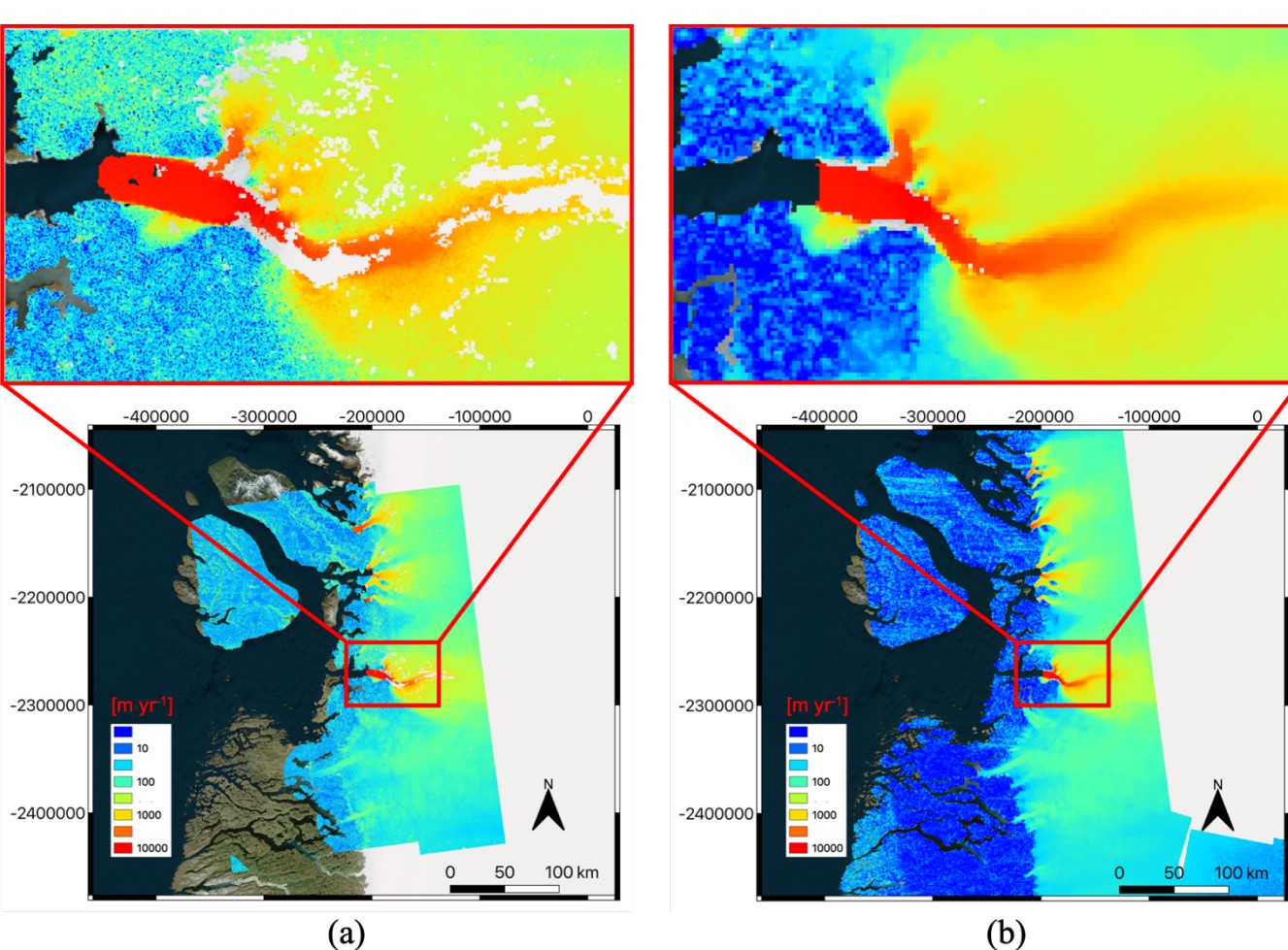

**Figure 12: Comparison of (a) a single 120 m posting 6-day ITS_LIVE Sentinel-1 image pair (acquired on 20170404-20170410) to (b) the PROMICE 500 m posting 24-day averaged (20170323-20170416) product (b) for Jakobshavn Isbræ Glacier in Greenland.**

We next compare time-series generated from the ITS_LIVE Sentinel-1 image pair product, the PROMICE product, and other
ITS_LIVE Version 2 image pair products acquired from Landsat-8 and Sentinel-2 sensors. The time series results and the error
metrics are shown in Fig. 13, where all of the times-series data were extracted at the location of the red cross marker (N
69.124°, W 49.496°) in Fig. 11. It is shown in Fig. 13 that all of the products generally agree well, capturing the large
interannual and seasonal velocity variation with a mean velocity around 10 km/year and an overall dynamic range of 8 km/year.
The error metrics are summarized as such: $R^2$ of 0.97 and Root Mean Square Error (RMSE) of 314 m/yr (relative percentage
of 3.7%) between ITS_LIVE Sentinel-1 and ITS_LIVE Version 2 Landsat 8; $R^2$ of 0.97 and RMSE of 354 m/yr (relative
percentage of 4.0%) between ITS_LIVE Sentinel-1 and ITS_LIVE Version 2 Sentinel-2; $R^2$ of 0.95 and RMSE of 512 m/yr
(relative percentage of 5.9%) between ITS_LIVE Sentinel-1 and PROMICE. In generating the above error metrics, all of the
other data products were re-sampled using the nearest neighbour method to the concurrent ITS_LIVE Sentinel-1 horizontal
(time) axis, where 695 ITS_LIVE Sentinel-1, 534 ITS_LIVE Version 2 Landsat 8, 1380 ITS_LIVE Version 2 Sentinel-2 image
pair products as well as 157 PROMICE products were used. Note for each year of 2016-2022, there are periods that ITS_LIVE
optical data are unavailable, which strengthens the adding value and competitive edge of using ITS_LIVE SAR data (currently
Sentinel-1).

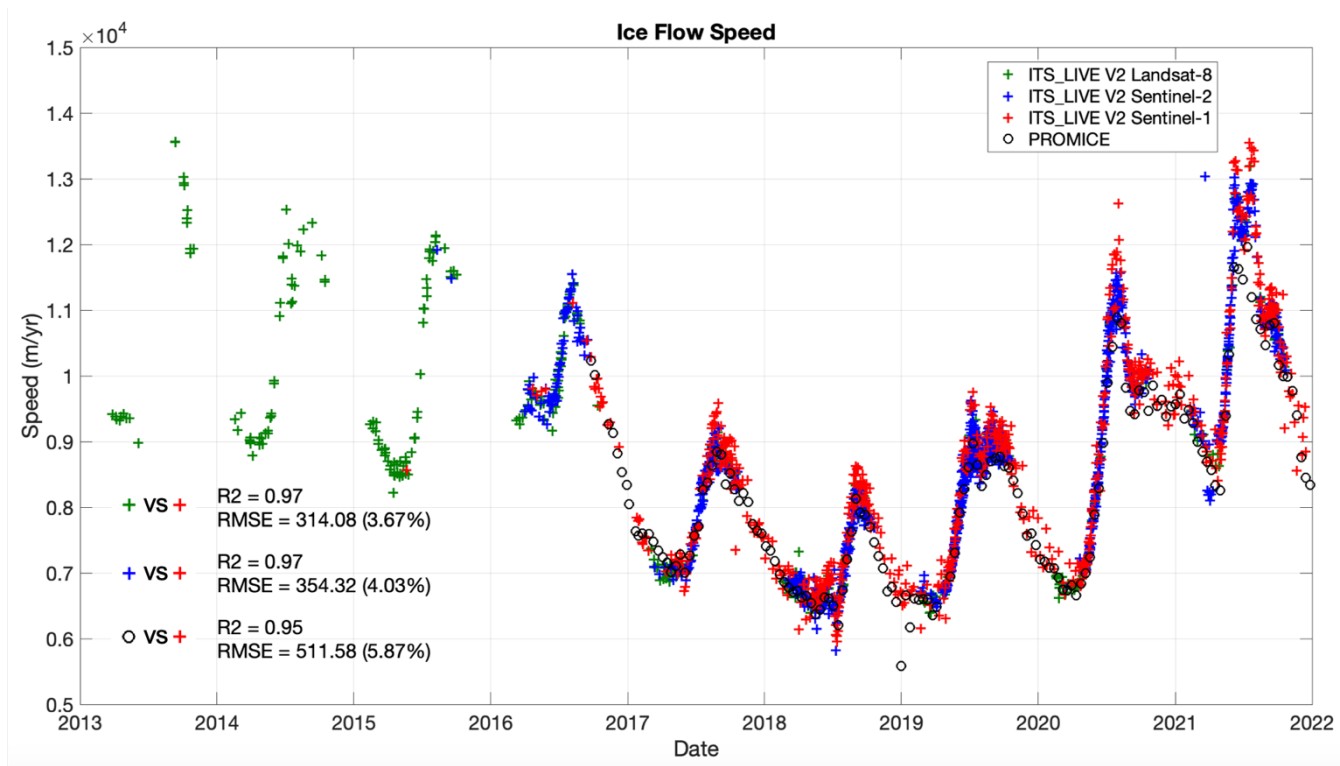

**Figure 13: Time series of ITS_LIVE Sentinel-1 image pair velocities, PROMICE velocities, and ITS_LIVE Version 2 Sentienl-2 and
Landsat-8 products for the Jakobshavn Isbræ Glacier in Greenland at the location of the red cross marker (N 69.124°, W 49.496°)
in Fig. 11. The inter-product comparison metrics are also shown. The comparison includes 695 ITS_LIVE Sentinel-1, 534 ITS_LIVE
Version 2 Landsat 8, 1380 ITS_LIVE Version 2 Sentinel-2 image pair velocities as well as 157 PROMICE velocities. The time**

separation between repeat images ranges between 16 to 544 days for Landsat 8 and 5 to 345 days for Sentinel 2 products. Note for each year of 2016-2022, there are periods that ITS_LIVE optical data are unavailable, which strengthens the adding value and competitive edge of using ITS_LIVE SAR data (currently Sentinel-1).

The **vx**, **vy**, and **v** uncertainty metrics of the ITS_LIVE Version 2 Sentinel-1 image-pair products shown in Fig. 13 are shown in Fig. 14 and are calculated using Eq. (3) as described at the end of Section 3.1.2. It is shown that an average uncertainty of 61 m/yr can be achieved while the error in **vx** (35 m/yr) is smaller than that in **vy** (71 m/yr) because of the coarser resolution in radar azimuth direction that mostly aligns with (thus impacts) the y-direction velocity. In generating the uncertainty metrics in Fig. 14, about 30% of the 695 ITS_LIVE Sentinel-1 image-pair products that were exploited in Fig. 13 were not used here due to the known issue of wrong identification of stable surfaces over fast-flowing Jakobshavn glacier outlet (flowing at a speed on the order of km/yr) when the valid pixels encompass the glacier outlet (with big ROI), which can in turn bias the uncertainty metric by a few hundreds of meters per year. The alternative approach of calculating the uncertainty by using the slowest 25% reference velocities as "stable surfaces" (Section 3.1.2) also tend to be problematic over such fast flow areas. Therefore, over fast-flowing glacierized regions, an accurate stable surface mask is required.

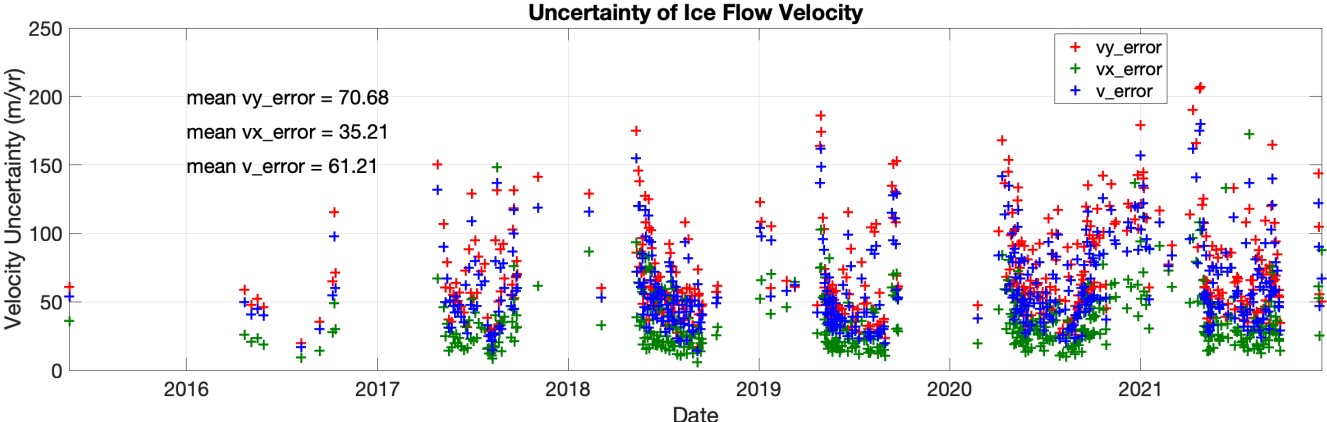

**Figure 14: Uncertainty in ITS_LIVE Sentinel-1 image pair velocities shown in Fig. 13 as provided in the product. Uncertainties of vx, vy and v are calculated according to Eq. (3).**

## 4.2 Pine Island Glacier

For validation over the Antarctic, we compare ITS_LIVE Sentinel-1 image pair products over the Pine Island Glacier (S 75°, W 100°; Fig. 15) to ITS_LIVE Version 2 Landsat-8 and Sentinel-2 image pair products, and to MEaSUREs Annual Antarctic Ice Velocity Maps, Version 1 (Mouginot et al., 2017b; Mouginot et al., 2017a) that are generated using multiple SAR and optical satellite data (JAXA's ALOS, ESA's ENVISAT and Sentinel-1, CSA's RADARSAT-1 and RADARSAT-2, DLR's TerraSAR-X and TanDEM-X, as well as USGS's Landsat-8). Pine Island Glacier is the fastest thinning glacier in Antarctica and is responsible for about 20% of Antarctica Ice Sheet's mass loss (Favier et al., 2014). The glacier has thinned at an increasing rate over the past 40 years with the grounding line retreated by tens of kilometres (Rignot et al., 2008; Favier et al., 2014).

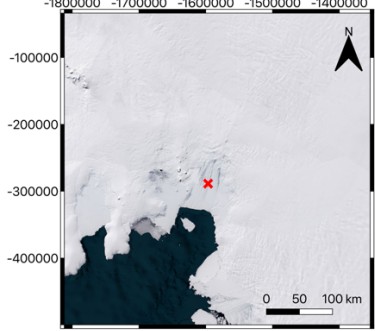

**Figure 15: Optical image of the Pine Island Glacier in Antarctic, where the red cross marks the location over the fast-moving glacier outlet that was used for the validation exercise.**

(a)

(b)

**Figure 16: Comparison of (a) a single 6-day (20190110-20190116) ITS_LIVE Sentinel-1 image pair velocity (b) with the MEaSUREs Annual Antarctic Ice Velocity Maps averaged for the period of July 2018 to June 2019 over the Pine Island Glacier in Antarctic.**

As a qualitative comparison, we show the ITS_LIVE Sentinel-1 image pair product (acquired on 20190110-20190116 and 20190122-20190128) along with the MEaSUREs annual ice velocity product in Fig. 16. Since the MEaSUREs Annual Antarctic Ice Velocity Maps have a temporal resolution of one year, we compare our data to the 2018-2019 mapping that is temporally averaged between July 2018 and June 2019. The difference in grid spacing can be noticed between the ITS_LIVE Version 2 product (grid spacing of 120 m) and the MEaSUREs annual product (grid spacing of 1000 m). It can also be seen from Fig. 16 that the MEaSUREs Annual Antarctic Ice Velocity Map has been smoothed spatially. The annual product is substantially better due to a significantly greater volume of data included in the annual average. However, the two products visually compare very well without noticeable difference for both slow- and fast-moving velocities.

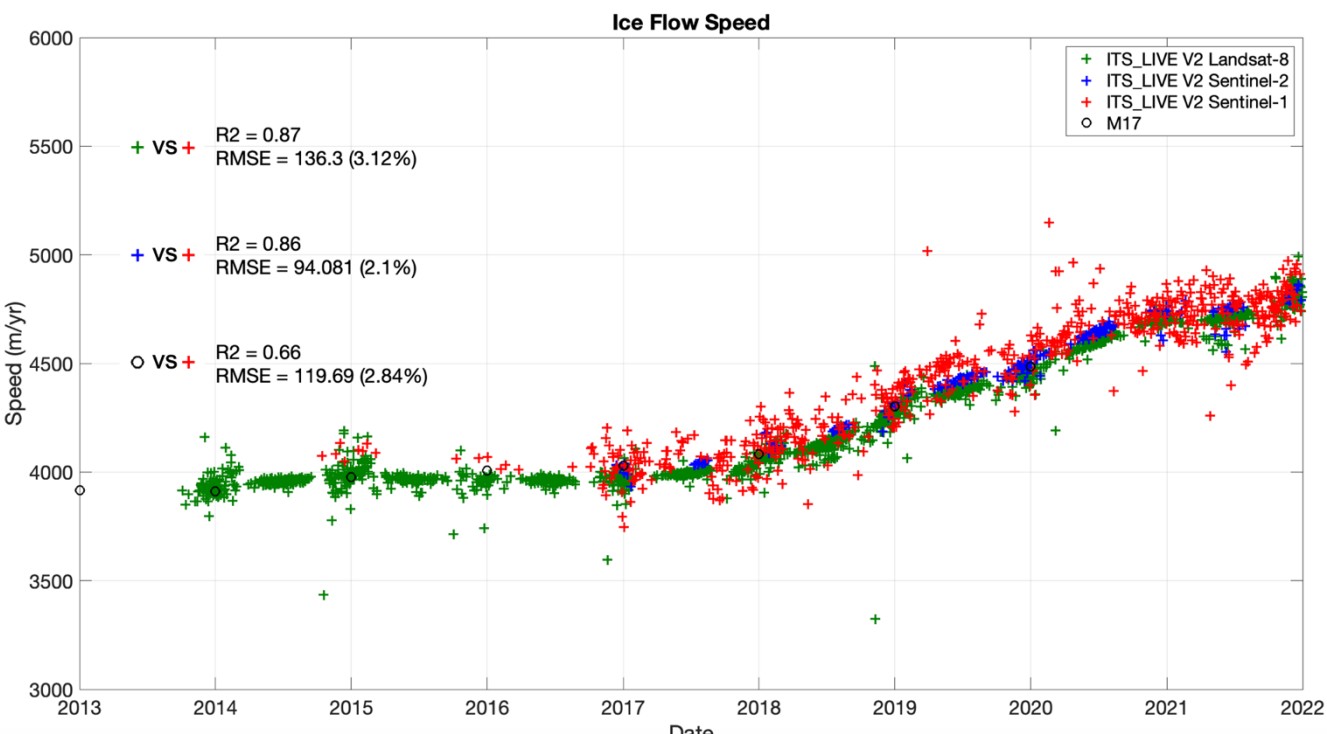

**Figure 17: Time series of ITS_LIVE Sentinel-1 image pair velocities, MEaSUREs Annual Antarctic Ice Velocity Maps (denoted as "M17"), and ITS_LIVE Version 2 Sentienl-2 and Landsat-8 products for Antarctica's Pine Island Glacier for the location of the red cross marker (S 75.14°, W 100.13°) in Fig. 15. The inter-product comparison metrics are also shown. The comparison includes 605 ITS_LIVE Sentinel-1, 1957 ITS_LIVE Version 2 Landsat 8, 223 ITS_LIVE Version 2 Sentinel-2 image pair products as well as 16 MEaSUREs Annual Antarctic Ice Velocity Maps. The time separation between repeat images ranges between 16 to 544 days for Landsat 8 and 10 to 530 days for Sentinel 2 products.**

Next, we show a cross-comparison between the ITS_LIVE Sentinel-1 image pair product, the MEaSUREs Annual Antarctic Ice Velocity Maps, and ITS_LIVE Version 2 Landsat-8 and Sentinel-2 products. The comparison and the error metrics are shown in Fig. 17, where all of the times-series data were extracted for the location shown by the red cross marker (S 75.14°,

W 100.13°) in Fig. 15. All of the products generally agree well, demonstrating both slow interannual and small seasonal velocity variation with the mean velocity around 4.35 km/year and an overall dynamic range of 750 m/year between 2013 and 2021. The error metrics are summarized as such: $R^2$ of 0.87 and RMSE of 136 m/yr (relative percentage of 3.1%) between ITS_LIVE Sentinel-1 and ITS_LIVE Version 2 Landsat 8; $R^2$ of 0.86 and RMSE of 94 m/yr (relative percentage of 2.1%) between ITS_LIVE Sentinel-1 and ITS_LIVE Version 2 Sentinel-2; $R^2$ of 0.66 and RMSE of 120 m/yr (relative percentage of 2.8%) between ITS_LIVE Sentinel-1 and the MEaSUREs Annual Antarctic Ice Velocity Maps. In generating the above error metrics, all data products were re-sampled using the nearest neighbour method to the concurrent ITS_LIVE Sentinel-1 horizontal (time) axis, where 605 ITS_LIVE Sentinel-1, 1957 ITS_LIVE Version 2 Landsat 8, 223 ITS_LIVE Version 2 Sentinel-2 image pair products, as well as 16 MEaSUREs Annual Antarctic Ice Velocity Maps, were used.

Similar to Section 4.1, **vx**, **vy**, and **v** uncertainty metrics of the ITS_LIVE Version 2 Sentinel-1 image-pair products used in Fig. 17 are shown in Fig. 18 and are calculated using Eq. (3). For the selected point, velocity magnitudes (**v**) has an average uncertainty of 67 m/yr while the uncertainty in **vx** (52 m/yr) is slightly smaller than that in **vy** (68 m/yr) due to the azimuth orientation that has slightly less projection onto the velocity in the x-direction for this location. Although Pine Island Glacier is on the other side of the planet from the Jakobshavn Isbræ Glacier (Section 4.1), the uncertainty metrics are very similar. This gives us confidence that the ITS_LIVE workflow produces consistent products for diverse regions of the globe.

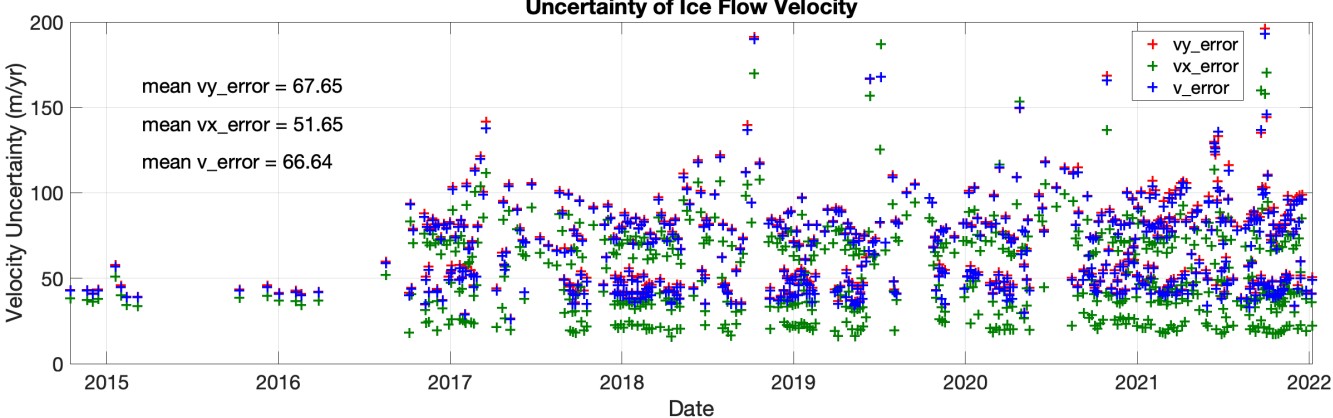

**Figure 18: Uncertainty in ITS_LIVE Sentinel-1 image pair velocities shown in Fig. 17 as provided in the product. The vx, vy and v uncertainties are calculated according to Eq. (3).**

## 4.3 Malaspina Glacier

Last, we validate the ITS_LIVE Sentinel-1 image pair products for a mountain glacier—Malaspina Glacier (N 60°, W 140°; Fig. 19), located in south-eastern Alaska, using both ITS_LIVE Version 2 image pair products acquired from optical sensors (Landsat-8 and Sentinel-2) as well as the FAU image-pair ice velocity products that are globally available and were also created from Sentinel-1 data via the offset-tracking method (Friedl et al., 2021). Malaspina glacier is the largest piedmont glacier in

the world, which also serves as a good test case for validating the described methodology and the image-pair products over mountainous areas with high topographic relief.

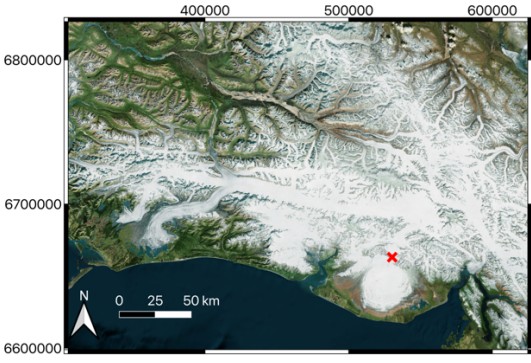

**Figure 19: Optical image of the test site at the Malaspina Glacier in Alaska, where the red cross marks the location over the fast-moving glacier outlet for the validation of the velocity estimates against other products.**

(a)                                                                                     (b)

**Figure 20: Comparison of the (a) ITS_LIVE Sentinel-1 image pair product to the (b) FAU product over the test site at the Malaspina Glacier in Alaska. Both datasets are generated using Sentinel-1 images acquired on 20190225 and 20190303.**

As a qualitative comparison, we show the ITS_LIVE Sentinel-1 image pair product (acquired on 20190225-20190303) alongside the FAU product in Fig. 20. FAU provides both monthly and annual mosaics and image-pair products with a temporal baseline of 6 and 12 days. For this comparison we were able to locate the FAU product that uses the same Sentinel-1 images

as the ITS_LIVE product. The difference in grid spacing and effective spatial resolution can be noticed between our ITS_LIVE Version 2 product (grid spacing of 120 m; effective spatial resolution of 240 m to 1920 m) and the FAU image-pair product (grid spacing of 200 m; effective spatial resolution of 800 m to 900 m). The FAU product appears to be smoothed/filtered spatially. This is due to FAU using a larger template window in offset tracking (thus suppressing some of the high-resolution details). Looking at the site overview subfigure of Fig. 20a and Fig. 20b, the FAU product also has less valid pixels over land

but about the same coverage over glacier surfaces, when compared to the ITS_LIVE product. Otherwise, both slow- and fast-flowing velocity estimates visually compare very well between ITS_LIVE Sentinel-1 and FAU image-pair products.

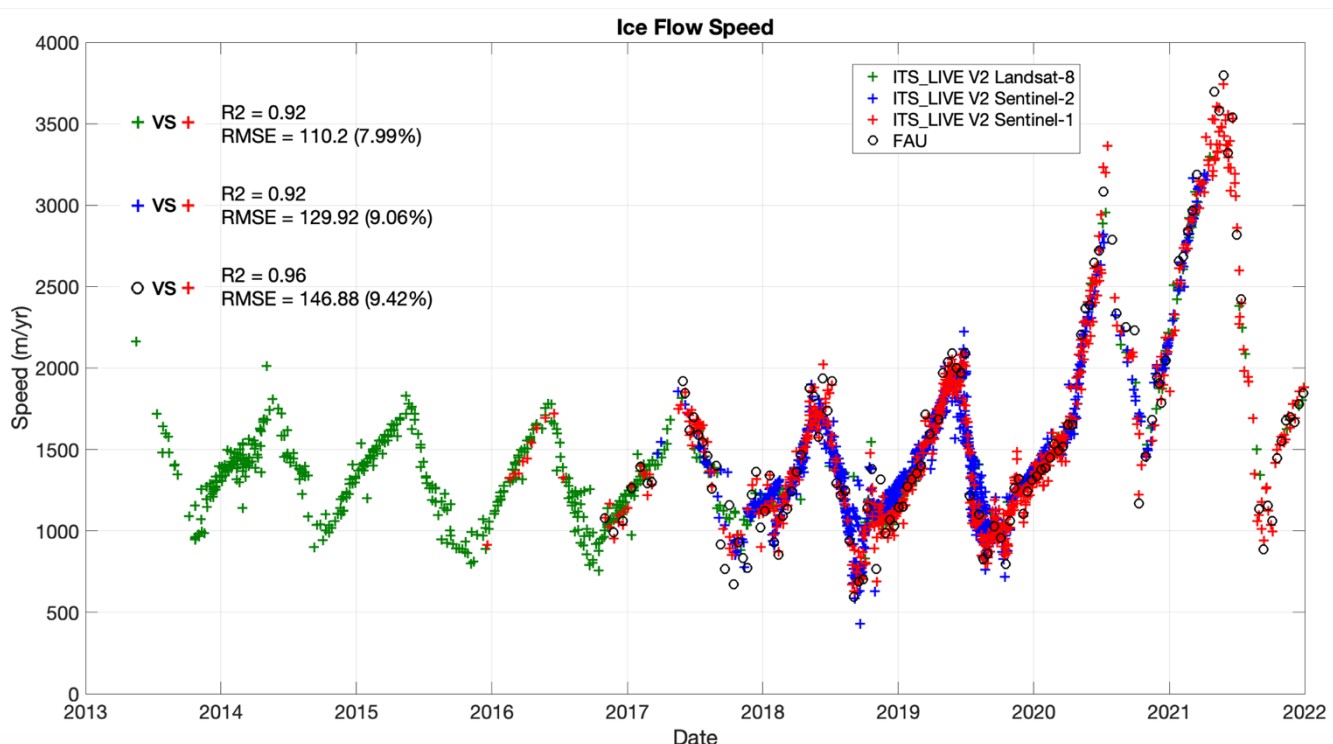

**Figure 21: Time series of ITS_LIVE Sentinel-1 image pair velocities, FAU velocities, and ITS_LIVE Version 2 Sentienl-2 and Landsat-8 products for Alaska's Malaspina Glacier for the location of the red cross marker (N 60.08°, W 140.47°) in Fig. 19. The inter-product comparison metrics are also shown. The comparison includes 715 ITS_LIVE Sentinel-1, 653 ITS_LIVE Version 2 Landsat 8, 1131 ITS_LIVE Version 2 Sentinel-2 image pair products as well as 134 FAU velocities. The time separation between repeat images ranges between 16 to 288 days for Landsat 8 and 5 to 280 days for Sentinel 2 products.**

Next, we show the comparison between the ITS_LIVE Sentinel-1, Landsat-8 and Sentinel-2 image pair products and the FAU
product. The time series results and the error metrics between various products are shown in Fig. 21, where all of the times-
series data were extracted at the location of the red cross marker (N 60.08°, W 140.47°) in Fig. 19. It is shown in Fig. 21 that
all of the products generally agree well, capturing the large interannual and seasonal velocity variation with the mean velocity
around 2 km/year and an overall dynamic range of 3 km/year, as well as a glacier surge after 2020. The error metrics are
summarized as such: $R^2$ of 0.92 and RMSE of 110 m/yr (relative percentage of 8.0%) between ITS_LIVE Sentinel-1 and
ITS_LIVE Version 2 Landsat 8; $R^2$ of 0.92 and RMSE of 130 m/yr (relative percentage of 9.1%) between ITS_LIVE Sentinel-
1 and ITS_LIVE Version 2 Sentinel-2; $R^2$ of 0.96 and RMSE of 147 m/yr (relative percentage of 9.4%) between ITS_LIVE
Sentinel-1 and FAU. In generating the above error metrics, all of the other data products were re-sampled using the nearest
neighbour method to the concurrent ITS_LIVE Sentinel-1 horizontal (time) axis, where 715 ITS_LIVE Sentinel-1, 653
ITS_LIVE Version 2 Landsat 8, 1131 ITS_LIVE Version 2 Sentinel-2 image pair products as well as 134 FAU products were
used.

Similar to Section 4.1 and Section 4.2, **vx**, **vy**, and **v** uncertainty metrics of the ITS_LIVE Version 2 Sentinel-1 image-pair
products shown in Fig. 21 are shown in Fig. 22 and are calculated using Eq. (3). It is shown that an average uncertainty of 58
m/yr can be achieved while the error in **vx** (51 m/yr) is slightly smaller than that in **vy** (61 m/yr) due to the azimuth orientation.
The uncertainty metrics for this mountain glacier are similar to those reported for Greenland and the Antarctic.

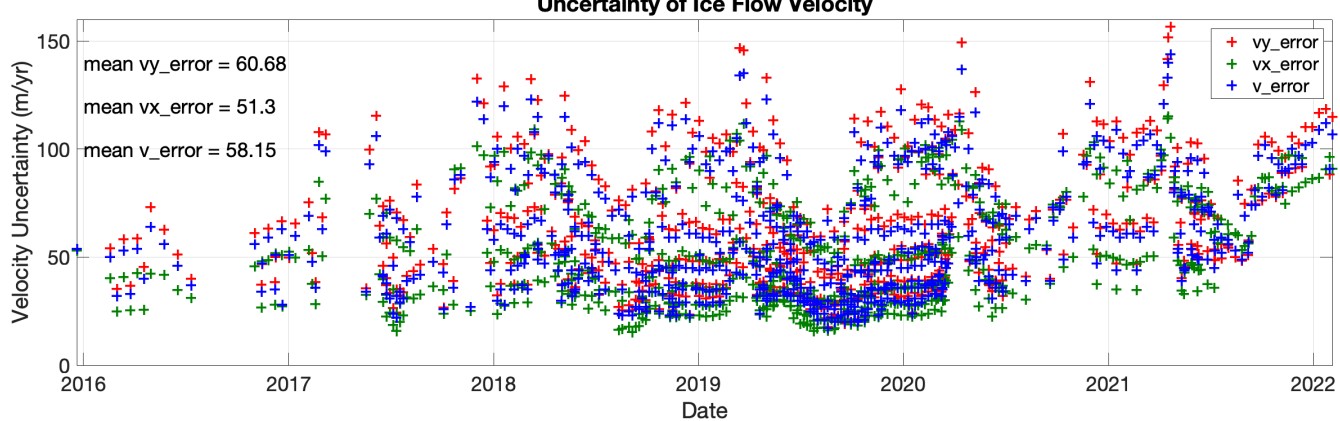

**Figure 22: Uncertainty in ITS_LIVE Sentinel-1 image pair velocities shown in Fig. 21 as provided in the product. The vx, vy and v uncertainties are calculated according to Eq. (3).**

Below we provide the implications of the data quality over smaller mountain glaciers. For global processing, autoRIFT uses
square search chip sizes ranging 240 m × 240 m to 1920 m × 1920 m. Only a single velocity vector is returned for a single
search chip. This means that when a single search chip covers a surface with steep spatial gradients in surface velocity (e.g.
shear margins, glacier margins, nunataks), only a single velocity vector will be returned. The returned vector represents the
displacement between features that provide maximum correlation. Rock often dominates the correlation for mixed search chips

that contain rock. This can cause glacier velocities to be negatively biased for narrow valley glacier and along shear margins. This same issue occurs for features that advect with the glacier (e.g. medial moraines) but present as stationary features. When a search chip samples these features the correlation can be dominated by the advecting moraine that appears as stationary. Lastly, Sentinel-1 is a side-looking SAR that is impacted by layover (i.e., multiple targets at the same range distance from the sensor are overlaid with each other causing their velocities mixed together) and line-of-sight shadowing (i.e., no targets appear at the side of a high mountain glacier facing away from radar resulting in missing velocity data) effects. Both of these issues are magnified in areas of high-relief where mountain glaciers are often located.

## 5 Summary and Conclusion

In this paper we describe the MEaSUREs ITS_LIVE Sentinel-1 Image-Pair Glacier and Ice Sheet Surface Velocities: Version 2 and the associated data processing algorithms. The dataset is provided for all large glacierized regions. To demonstrate the data quality and algorithms we showcase 21 Sentinel-1 image-pairs (Lei et al., 2021b) over three different sites in Greenland: one covering the Jakobshavn Isbræ Glacier to demonstrate algorithm performance in an area of fast flow and two other locations covering the interior of the ice sheet for calibration purposes.

We first summarized the ITS_LIVE Sentinel-1 image pair product along with the data processing chain. Both the inputs and outputs of the processing chain are introduced in detail. These image pair products are generated by using the offset tracking processing chain that consists of two Python modules: *Geogrid* and *autoRIFT*. In this work, the core offset tracking module, *autoRIFT*, has been enhanced with a few techniques for better accuracy, processing efficiency and to accommodate overlapping search chips for finer resolution. Benchmark comparisons between *autoRIFT* and ampcor show *autoRIFT* achieves multiple orders of magnitude improvement in efficiency, while at the same time providing a ~33% reduction in uncertainty.

We then demonstrate approaches to reduce systematic error in Sentinel-1 image pair products. First, the slant-range/azimuth offset tracking results are calibrated for the subswath and full-swath dependent geolocation bias for Sentinel-1A/B pairs. By investigating 11 pairs, the average pixel offset bias between subswath 2 and 1 are -0.010 pixel in slant-range and 0.019 pixel in azimuth, while the bias between subswath 3 and 2 are -0.007 pixel in slant-range and 0.006 pixel in azimuth, which is referenced to an image pair with Sentinel-1A acquired prior to Sentinel-1B. These subswath offset bias estimates are then used as a static correction of the subswath dependent geolocation error in each ITS_LIVE Sentinel-1 image pair product. Following this, the full-swath dependent geolocation bias is corrected in a dynamic way for each image pair by using stable surfaces (ice-free terrain and slow-moving ice <15 m yr$^{-1}$). Averaging 20 pairs, the calibrated velocity bias is -4.1 m yr$^{-1}$ in slant range and -26.8 m yr$^{-1}$ in azimuth, which is referenced to a 6-day Sentinel-1A/B pair with Sentinel-1A acquired prior to Sentinel-1B. A secondary method is tested with almost equivalent performance that uses the area of the slowest 25% of the reference velocity whenever the image pair does not intersect the stable surface. Even in well calibrated constellation missions, instrument-level differences that result in full-swath or subswath level offsets can exist. When Sentinel-1C and Sentinel-1D are launched,

similar offset calibration exercises are needed before offsets generated from their imagery can be incorporated into ITS_LIVE like projects and/or processing campaigns.

To reduce the impact of ionosphere induced azimuth streaks, the product includes LOS parameters that support x/y horizontal velocity inversion from two slant-range measurements of both ascending and descending geometry. Using LOS measurements from two Sentinel-1 image pairs (one ascending and one descending) with 6-day separation, as well as the LOS parameter layers in the ITS_LIVE products, the uncertainty reduced from 8.6 m yr$^{-1}$ (ascending) or 3.7 m yr$^{-1}$ (descending) to 3.4 m yr$^{-1}$

(by 9-61%) in the x-direction and from 12.9 m yr$^{-1}$ (ascending) or 10.9 (descending) to 5.6 m yr$^{-1}$ (by 49-57%) in the y-direction. After applying the ionosphere correction approach in this paper, there is still some residual slow-varying trend due to ionosphere disturbance in the LOS offset tracking velocity estimates, which could be removed by referencing to known ground control points with zero velocity through a quadratic baseline fit (Mouginot et al., 2012; Joughin et al., 2017). As for future work, we plan to investigate the possibility of using split-spectrum InSAR phase (Liang et al., 2019) to remove the slow-

varying ionosphere trend from the SAR LOS offset-tracking result over high-coherence areas, and then using the ionosphere correction approach (Section 3.2) in this paper to derive velocity products solely from corrected SAR LOS measurements, which could help to mitigate the artifacts from ionosphere-caused slow-varying gradients (as well as azimuth streaks).

We further validated the ITS_LIVE Version 2 Sentinel-1 image pair products (with 6-year time series of thousands of epochs)

over three test sites covering the globe: the Jakobshavn Isbræ Glacier in Greenland, the Pine Island Glacier in Antarctica, and the Malaspina Glacier in Alaska. Comparing with other similar products (PROMICE, FAU and MEaSUREs Annual Antarctic Ice Velocity Maps) and ITS_LIVE Version 2 products from optical sensors (Landsat-8 and Sentinel-2), we find the overall deviation between products around 100 m/yr over fast-flowing glacier outlets (where both mean velocity and dynamic variation are on the order of km/yr) and increases up to 300-500 m/yr (3-6%) for the fastest Jakobshavn Isbræ Glacier. The uncertainty

of the products has been shown to be uniformly distributed around 60 m/yr for the velocity magnitude for the three regions investigated.

Other limitations and future work entail further improvements of the dynamic geolocation bias calibration when the stable surface mask is unavailable within the spatial extent of the image pairs, and ionosphere-induced azimuth streak removal by

using an a-priori flow direction. The approaches presented here are directly applicable to future radar satellite missions (e.g. NASA-ISRO's NISAR). It is our hope that by providing state-of-the-art, low latency, glacier velocity products to the public we will accelerate the understanding of glacier and ice sheet response to changes in ocean and atmosphere.

## 6 Data availability

The sample ITS_LIVE Sentinel-1 image pair products for the 21 Sentinel-1 image pairs used for methodology demonstration

in this paper can be found at the following DOI: https://doi.org/10.5281/zenodo.5606118 (Lei et al., 2021b). The final release of MEaSUREs ITS_LIVE Sentinel-1 Image-Pair Glacier and Ice Sheet Surface Velocities: Version 2 ice velocity product

(including image pair maps, datacubes and mosaics) for Sentinel-1 (https://doi.org/10.5067/0506KQLS6512) as well as other optical sensors (Landsat-4/5/6/7/8 and Sentinel-2) can be found at the ITS_LIVE project website: https://its-live.jpl.nasa.gov. Regarding the input files for the global processing, we use the Copernicus DEM GLO-30 (https://spacedata.copernicus.eu/web/cscda/dataset-details?articleId=394198) and ITS_LIVE Version 1 Landsat-8-derived velocity mosaics (https://its-live.jpl.nasa.gov) as the reference velocities for deriving all our ITS_LIVE Version 2 products. Regarding the cross-validation data, the PROMICE ice velocity products can be downloaded from https://doi.org/10.22008/promice/data/sentinel1icevelocity/greenlandicesheet. The FAU image pair products can be found at http://retreat.geographie.uni-erlangen.de. The MEaSUREs Annual Antarctic Ice Velocity Maps can be found at https://nsidc.org/data/NSIDC-0720/versions/1.

## 7 Software Tools

The MEaSUREs ITS_LIVE Sentinel-1 Image-Pair Glacier and Ice Sheet Surface Velocities: Version 2 are processed with the following software tools. First, ISCE Version 2.4+ (https://github.com/isce-framework/isce2/releases; in particular the "topsApp" function) is used to preprocess the two Sentinel-1 images that form an image pair up to the step of "mergeBursts", which results in co-registered SLC images. Then, *Geogrid*/*autoRIFT* Version 1.4.0 (https://doi.org/10.5281/zenodo.5643820) is used to generate the final NetCDF data product. For more details, the readers are referred to the help page of ISCE (https://github.com/isce-framework/isce2), *Geogrid* (https://github.com/leiyangleon/Geogrid) and *autoRIFT* (https://github.com/nasa-jpl/autoRIFT).

*Author contributions.* AG conceived the ITS_LIVE project. AG and PA provided the high-level designs of the Sentinel-1 ice velocity processing and data product. YL, AG and PA developed the processing software (*autoRIFT*/*Geogrid*), designed cal/val tests, and set up the format and layers of the final NetCDF product. YL prepared the manuscript with contributions from all co-authors.

*Competing interests.* Authors declare that they have no conflict of interest.

*Acknowledgement.* This effort was funded by the NASA MEaSUREs program in contribution to the Inter-mission Time Series of Land Ice Velocity and Elevation (ITS_LIVE) project (https://its-live.jpl.nasa.gov) and through Alex Gardner's participation in the NASA NISAR Science Team.

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
