# Peer review of "Processing methodology for the ITS\_LIVE Sentinel-1 ice velocity product"

_Earth System Science Data, 2021_

## Author Comment (AC1)

We thank the topical editor (TE) and the two reviewers for the constructive comments and suggestions that indeed improve the manuscript a lot. The paper has been heavily revised by rewriting and adding several sections, e.g. the entire new section (Section 4) on global cross-validation as pointed out by all of the TE and the two reviewers. Here we first summarize the major changes in this revision:

1. An entire new section (the new Section 4) has been added to validate the products over three typical sites covering the globe: Jakobshavn Glacier in Greenland, Pine Island Glacier in Antarctic, and Malaspina Glacier in Alaska. As cross-validation of the ITS_LIVE Sentinel-1 products, we compare with other similar products (PROMICE, FAU and other MEaSUREs data) as well as other ITS_LIVE Version 2 optical products (from Landsat-8 and Sentinel-2);
2. Section 2.1.2 on the description of data product has been revised a lot by including more details of the product and how to use it;
3. Section 1 (Introduction) has been rewritten to incorporate more literature review on glaciological use of velocity, other methods for deriving velocity, and other similar products, etc;
4. Abstract has been heavily revised to include more technical details;
5. Section 2.1.1 (Input dataset) has added descriptions of the input files (DEM and reference velocity) for global processing;
6. Two new figures are added, namely, Figure 2 (global view of the data availability) and Figure 3 (workflow of autoRIFT);
7. Uncertainty discussion is elaborated at the end of Section 3.1.2 and in the newly added Section 4 (cross-validation);
8. The new Section 6 (Data availability) has added a lot of new materials on the dataset used.

Below we also addressed all of the TE's and both reviewers' comments in an item-by-item manner. Thanks very much for your favorable consideration of this work.

Sincerely,

Yang Lei

on behalf of the authors

**Topical Editor**

Dear Authors,

In addition to the two reviewer comments, I have the following observations, suggestions, comments, and questions.

The current version only validates against other Sentinel SAR imagery. I think it would strengthen the data description to take some outputs and compare against one of the many already existing velocity products. See Sect. 3.5 of https://essd.copernicus.org/articles/10/2275/2018/ I suggest validating against one or more of the many other MEaSUREs (or PROMICE) velocity products in Greenland, and other velocity products in Antarctica and in alpine regions. Why should I use this instead of the many other MEaSUREs or PROMICE products? How does it compare to them? What are the causes of the disagreements, assuming they exist? Are the errors random or systematic bias?

We thank the TE for this constructive comment. In response to the TE's and both reviewers' comment, we have added an entire new Section 4 for the cross-validation with other similar products, e.g. PROMICE, FAU and other MEaSUREs dataset as well as other ITS_LIVE Version 2 optical products (from Landsat-8 and Sentinel-2). Please refer to the major change #1 listed on the front cover of this response.

All errors appear internal to your processing scheme. Are there other errors that impact the final data product? See Sect. 3.6 of above URL.

Both Abstract and Section 2.3 discuss the measurement uncertainty (x/y pixel offset precision) inherent to Sentinel-1 data itself. Section 3 discusses geolocation bias and ionosphere effects along with correction methods, which are either systematic to Sentinel-1 data or external error sources. After applying all of the corrections proposed in this work, the resulting uncertainty of the final product is around 60 m/yr worldwide as shown in the newly added cross-validation section (Section 4). To account for the 60 m/yr uncertainty, no other errors have yet been found to have a substantial contribution.

Given ITS_LIVE claim of global, only 3 validation regions in Greenland is limiting. Antarctica and especially small Mountain Glaciers are fundamentally different creatures - are the results and errors the same in those locations? I think an in-depth discussion of your data product and its quality issues is needed for mountain glaciers and Antarctica. Reviewer 1 suggests that the paper title could be changed to reflect "polar regions", but I disagree. The dataset is global, so this data description paper must cover the data, and cannot focus on just a subset of the data.

We thank the TE for this comment on global validation and not changing the title to polar regions. In response to the TE's and both reviewers' comment, we have added an entire new Section 4 for the cross-validation with other similar products over three typical sites covering the globe: Jakobshavn Glacier in Greenland, Pine Island Glacier in Antarctic, and

Malaspina Glacier in Alaska. Please refer to the major change #1 listed on the front cover of this response.

A more detailed description of the data product is needed. For example, how many images are there? How many have 6 day resolution, and how many have 60 day resolution? Are there differences between A and B? Which should I use for my (insert common use case scenario here)? Regarding the Greene et al (2020) paper, how does that impact the 6 vs 60 day w.r.t. under/over sampling highs/lows? By "large glacierized regions" are you really "global" or do you exclude small mountain glaciers? What criteria defines your cutoff?

Please refer to the major change #2 listed at the front cover of this response. The relevant text (line # 196-208) is also copied here:

> *Image pair products span the time period from the launch date of the corresponding sensors (e.g., Sentinel-1A—Sentinel-1A image pairs are available after the Sentinel-1A launch date, Sentinel-1B—Sentinel-1B and Sentinel-1A—Sentinel-1B or Sentinel-1B—Sentinel-1A image pairs are available after the launch date of Sentinel-1B) to present with a project goal of providing < 3 months latency from the date of acquisition. The separation times for the image pairs vary from 6 to 12 days for all sensor combinations, with the goal of extending to 60 days with multiples of 6-day separation dependent on funding. As a rule of thumb, for working with individual image-pair products over fast-flowing glacierized regions where rapid temporal decorrelation of radar signals is expected, using shorter separation times if available (e.g. 6 day; Sentinel-1A—Sentinel-1B or equivalently Sentinel-1B—Sentinel-1A) usually has better data quality than longer ones (e.g. 12 days or larger; Sentinel-1A—Sentinel-1A or Sentinel-1B—Sentinel-1B) due to less temporal decorrelation, which has been investigated by (Friedl et al., 2021; Solgaard et al., 2021). However, for observing slow-moving targets, all image-pair products with various separation times need to be considered to fully capture both the short-term and long-term variability of the velocities, such as in studying the seasonal ice dynamics (Greene et al., 2020).*

Since ITS_LIVE image-pair product is an operational product with new data coming in as time goes by, it is not necessary to count how many images and/or how many 6 days vs. 12 days pairs there are, as these numbers are temporal varying. For example, other similar products (PROMICE and FAU that were published by ESSD) do not provide such information either. We thus appreciate the TE's understanding on this.

Currently we include all glacier regions that contain glaciers larger than about 25 km$^2$, which is now emphasized in the revised text, as pointed by the TE.

Ken Mankoff

Editor

**Reviewer #1**

This paper presents a detailed methodology of processing Sentinel-1 radar data (TOPS mode) using an updated module "autoRIFT '' of ISCE platform in order to generate ITS_LIVE Sentinel-1 ice velocity products. The paper walks through the different elements of the sequential processing chain and highlights key points that improve the resolution and accuracy of the velocity products. The paper is clear and generally well-written. I have a few comments and suggestions which may be incorporated for more clarity.

We thank the reviewer for the recognition of this work.

**MAJOR COMMENTS:**

1. The products will be openly available. The module of ISCE platform will also be publicly available. These free resources will be used by many folks across the globe for their scientific analysis or processing data over areas other than polar regions. This paper limits the presentation and analysis of ITS_LIVE velocity products to Greenland, but the products will be available for mountain glaciers as well. The title should reflect this aspect; maybe by adding "polar regions '' in the title. Alternatively, more insights based on presentation and analysis over mountainous regions (e.g. European Alps, High-Asia) should be added in the paper. It is obvious that ITS_LIVE products and associated uncertainties are different in regions other than polar ice sheets.

We thank the reviewer for this constructive comment. In response to the TE's and both reviewers' comment, we have added an entire new Section 4 for the cross-validation with other similar products over three typical sites covering the globe: Jakobshavn Glacier in Greenland, Pine Island Glacier in Antarctic, and Malaspina Glacier in Alaska (mountain glacier). Please refer to the major change #1 listed on the front cover of this response.

2. We have a number of ice velocity products based on Sentinel-1 radar data and it is increasingly challenging which product is the best way to carry out a scientific analysis without processing the raw GRD/SLC data. Boncori et al., 2018 compared ice velocity products from several international research groups, highlighted different strategies on the processing and uncertainty estimation and found significant differences, also recommending a universal approach. This paper provides a new or updated algorithm (which is great) but needs to be compared with similar contemporaneous products (e.g. PROMICE). Otherwise, the scientific users will have to do this exercise or cherry-pick one of the available products. Both will not serve the ongoing efforts of establishing standard method development, ice velocity product generation and documentation. It would be nice to compare Sentinel-1 ITS_LIVE products with previous ITS_LIVE products obtained from optical remote sensing data (e.g. Landsat).

Thanks to the reviewer for this constructive comment as well. In the newly added Section 4, we have also included the cross-validation with other similar products, e.g. PROMICE, FAU and other MEaSUREs dataset as well as other ITS_LIVE Version 2 optical products

(from Landsat-8 and Sentinel-2). Please also refer to the major change #1 listed on the front cover of this response.

3. If this paper serves only a method development, there should be some more test cases (e.g. ice shelves in Antarctica, debris-covered glaciers in Alaska/high-Asia) to present the applicability of the algorithm other than Greenland Ice Sheet.

As mentioned in the major changes listed on the front cover of this response as well as in the revised text (abstract, introduction, etc), this paper provides the information of the data product, processing details and validation, not only the method. However, as shown above, the newly added Section 4 includes the cross-validation with other similar products over three typical sites covering the globe: Jakobshavn Glacier in Greenland, Pine Island Glacier in Antarctic, and Malaspina Glacier in Alaska. With this detailed time series analysis of velocity / uncertainty, it is concluded that the algorithm and products are applicable to other regions of the globe besides Greenland Ice Sheet.

4. So finally these products will not be average over a certain time-period like PROMICE 21-day ice velocity mosaics? Please clarify and highlight, if this is the case, in your paper as this is a unique aspect.

In response to the reviewer's comment, we added the following sentences (line # 71-77). Please also note that PROMICE is 24-day time averaged.

> *Among these efforts, the NASA MEaSUREs project Inter-mission Time Series of Land Ice Velocity and Elevation (ITS_LIVE) releases ice velocity products, i.e., 1) image-pair granules (without time averaging), 2) datacubes (time series of image-pair results) and 3) regional mosaics (averaged both spatially and temporally) with global coverage using temporally dense multi-sensor observations from both optical (Landsat 4/5/6/7/8 and Sentinel-2) and SAR (Sentinel-1) satellite data (Gardner et al., 2018). Other similar products of regional and/or global ice velocities have also been released, such as the Programme for Monitoring of the Greenland Ice Sheet (PROMICE) Ice Velocity product (Solgaard and Kusk, 2021) that releases temporally (24-day) averaged velocity mosaics over Greenland Ice Sheet*

**MINOR COMMENTS:**

L40: Several satellite derived regional ice velocity products are released annually

Fixed

L45: As described in Lei et al., 2021a (CHECK ELSEWHERE)

Fixed

L55: 6 days repeat is not everywhere but limited to polar regions and Europe or some key areas of the world.

It has been reworded as (line # 90-91)

> *with a repeat cycle of 6 days between A and B satellites over polar regions and some other key areas of the world.*

L70: Revise " We do …… Greenland"

It has been revised as below (line # 109)

> *Both inputs to and outputs of the processing chain are demonstrated for a set of test data collected over Greenland.*

L90: Have you ever considered 2m Arctic DEM instead of GIMP DEM? That may be a better choice for transformation between radar and geographic coordinates.

Thanks to the reviewer on suggesting other high-resolution DEMs. We have added a discussion about this (line # 128-134).

> *In Fig. 1b, we show the GIMP Digital Elevation Model (DEM) for the Greenland Ice Sheet (Howat et al., 2014; Howat et al., 2015), which is used in this work for illustration purposes only. Note for the global processing, we considered various DEM's with different resolution, e.g. Arctic DEM, REMA DEM, TanDEM-X DEM, Copernicus DEM, and NASADEM, and found the Copernicus DEM with global coverage at 30 m resolution (GLO-30) is the best available large-scale DEM which was also baselined for NASA-ISRO's NISAR mission after extensive analysis. With this analysis, it is found DEM's with varying resolutions have negligible effects on the resulting offset-tracking velocities given the grid spacing of 120 m used. Geolocation accuracy is however slightly more sensitive to the DEM resolution and accuracy.*

L95: There is no reference velocity for mountain glaciers. What will be the approach for those areas?

In response to the reviewer's comment, we also added the following discussion (line # 155-159).

> *As Fig. 1 shows, the above inputs of DEM and reference velocities are specifically chosen for Greenland Ice Sheet and for illustration purposes only. For the global processing, including Arctic, Antarctic and all the other areas of the world, e.g. high mountain glaciers, we use the Copernicus DEM GLO-30 (https://spacedata.copernicus.eu/web/cscda/dataset-details?articleId=394198) and ITS_LIVE Version 1 Landsat-derived velocity mosaics (https://its-live.jpl.nasa.gov) as the reference velocities for deriving all our Version 2 ITS_LIVE products.*

L100: What do you mean by "successful match"? Based on cross-correlation or similarity function value?

We have revised it as below (line # 143-145)

> *autoRIFT will cycle from the minimum to the maximum chip size until a "valid" offset is found (i.e., passing the NDC filter based on the derived pixel displacement similarity with adjacent chips; as discussed in Section 2.2)*

L130: ..the extent of..

Fixed

L140-145: Repetitive

Fixed

L175 or elsewhere: "autoRIFT" should be clearly distinguished – italic?

Fixed throughout the manuscript.

L180-200: I strongly recommend a graphical representation of an algorithm – clearly distinguishing chip size, overlapping region, search size etc. on images.

In response to the reviewer's comment, we have added the new Fig. 3 along with Fig. 4 for illustrating the workflow of autoRIFT. As mentioned in the text, the readers are also referred to the previous publications (e.g. Gardner et al., 2018 and Lei et al., 2021a) on more details of the algorithm, which is outside the scope of this paper that is focused on data product.

L220-230: It was not very much clear why it was done that way. A lot of parameters with equations break the flow of writing here. Simpler writing with rationale might help us better understand.

We understand the reviewer's concern. Since the steps with those parameters are relevant for introducing the implementation of the upgraded NDC filter, we appreciate the reviewer's understanding on keeping the steps as well as the parameters. However, in response to the comment, we have added a few words to clarify each step (line # 288-293).

> *b. Compute the normalized displacement disparity in reference to the central grid point for all the grid points within the filter window and count the number of grid points that have displacement disparity smaller than FracSearch (threshold of maximum disparity with the default value of 0.2);*
>
> *c. If the number of grid points in the above step is greater than or equal to FracValid $\times$ FiltWidth2 (by definition, it is the threshold of valid points within the filter window, e.g., $8/25\cdot5^2=8$ in this case), the central grid point is retained; otherwise, it is discarded;*

L290: Is this subjective? Any insights?

Results determined using an oversampling ratio of 1/128 are considered most precise in this work, which is approaching the limit of the computing resources regarding the trade-off of efficiency/accuracy.

L305: 7>>seven

Fixed

L450: Maybe I miss something, but the slant range displacement component contributes to ground range (East-West (your x) and North-South (your y)) and vertical movement (z) and the azimuth displacement component contributes to x and y only. Your equations don't consider vertical velocities. It is known that the vertical velocities exist as well due to slope/elevation changes in the flow direction or ablation. Please clarify.

According to this comment, we have added a few sentences to clarify it (line # 531-535). The slope parallel assumption (Joughin et al., 1998) has been applied here to constrain the vertical velocity.

As described in Section 2.1.3 and in Lei et al. (2021a), the Geogrid module provides a look-up table of $2 \times 2$ conversion matrix between range/azimuth pixel displacements and x-/y-direction velocity fields. Here, the slope parallel assumption (Joughin et al., 1998) is adopted, where surface displacement is assumed to be parallel to the local surface. Therefore, the z-direction (vertical) velocity can be further expressed as a function of the x-/y-direction velocities. Note this assumption becomes problematic over regions of known non-trivial vertical flow, e.g., slope and/or elevation changes.

**Reviewer #2**

This study provides a processing chain for Sentinel-1 TOPS mode data. Authors have used a modified module of autoRIFT to generate ice velocity global products. This paper is successful in demonstrating the processing chain and efforts in overcoming associated errors. This study also demonstrated improvement in terms of accuracies and resolution of velocity products. The paper is methodologically well organized. However, I have a few comments. Authors may include these comments from the reader's perspective. My sequential comments are;

Thanks to the reviewer for the recognition of this work.

Abstract: Authors should re-write the abstract clearly indicating quantitative improvements when they say *higher accuracy*, *finer resolution*, *improvements*. In the present abstract, the reader cannot find what level of improvements, how much accuracy, and what resolution authors refer to. In general, this abstract reads very generically as a project report. To attract a wider audience, this abstract should be heavily revised to clearly state main achievements quantitative, limitations of the current product, and comparative analysis with existing data.

Please refer to the major change #4 listed on the front cover of this response. In response to the reviewer's comment, we have thoroughly rewritten the abstract to reflect the actual numbers and technical details.

> *…*
>
> *We demonstrate improvements to the core processing algorithm for dense offset tracking, "autoRIFT", that provides finer resolution (120 m instead of the previous 240 m used for Version 1) and higher accuracy (20% to 50% improvement) data products with significantly enhanced computational efficiency (>2 orders of magnitude) when compared to earlier versions and the state-of-the-art "dense ampcor" routine in JPL's ISCE software. In particular, the disparity filter is upgraded for handling finer grid resolution with overlapping search chip sizes, and the oversampling ratio in the subpixel cross-correlation estimation is adaptively determined for Sentinel-1 data by matching the precision of the measured displacement based on the search chip size used.*
> *…*
>
> *After the proposed correction of ionosphere errors, the uncertainties in velocities are reduced by 9-61%. We further validate the ITS_LIVE Version 2 Sentinel-1 image pair products, with 6-year time series composed of thousands of epochs, over three typical test sites covering the globe: the Jakobshavn Isbræ Glacier of Greenland, Pine Island Glacier of Antarctic, and Malaspina Glacier of Alaska. By comparing with other similar products (PROMICE, FAU, and MEaSUREs Annual Antarctic Ice Velocity Map products), as well as other ITS_LIVE Version 2 products from Landsat-8 and Sentinel-2 data we find an overall variation between products is around 100 m/yr over fast-flowing glacier outlets, where both mean velocity and variation are on the order of km/yr, and increases up to 300-500 m/yr (3-6%) for the fastest Jakobshavn Isbræ Glacier. The velocity magnitude uncertainty of the ITS_LIVE Sentinel-1 products is calculated to be uniformly distributed around 60 m/yr for the three test regions investigated.*

Introduction: This section only focuses on operational velocity product generation attempts. Authors should enrich this section by providing literature on glacier velocity generation in general, the use of glacier velocity in glaciological studies, various ways of deriving velocity fields and uncertainties, and the strengths of different methods. Currently, this introduction section does not discuss existing regional attempts using a variety of

methods. After the literature review, the authors should provide gaps in current knowledge and what additional knowledge this study provides to the scientific community.

Please refer to the major change #3 listed on the front cover of this response. In response to the reviewer's comment, we have added more literature review on glaciological use of velocity, other methods for deriving velocity, and other similar products, etc.

2.1 Product and methodology overview

2.1.1 Input dataset: This section provides only example input datasets for the Greenland case study. I will suggest including other reference input datasets used for other regions. This can be added in the supplementary information as a table or description. Reference velocity for the Greenland ice sheet is mentioned in the study but I am wondering which stable reference velocities are being used for other parts of the globe. Similarly, input DEMS for other regions should be included in the supplementary information. The impact of varying resolution of DEMs in different regions should be discussed. Have you tried Arctic DEM?

Thanks to the reviewer for the comment on input files for global processing, which has been addressed by the major change #5 listed on the front cover of this response. In particular, we have added a discussion about them (line # 128-134).

> In Fig. 1b, we show the GIMP Digital Elevation Model (DEM) for the Greenland Ice Sheet (Howat et al., 2014; Howat et al., 2015), which is used in this work for illustration purposes only. Note for the global processing, we considered various DEM's with different resolution, e.g. Arctic DEM, REMA DEM, TanDEM-X DEM, Copernicus DEM, and NASADEM, and found the Copernicus DEM with global coverage at 30 m resolution (GLO-30) is the best available large-scale DEM which was also baselined for NASA-ISRO's NISAR mission after extensive analysis. With this analysis, it is found DEM's with varying resolutions have negligible effects on the resulting offset-tracking velocities given the grid spacing of 120 m used. Geolocation accuracy is however slightly more sensitive to the DEM resolution and accuracy.

and (line # 155-162)

> As Fig. 1 shows, the above inputs of DEM and reference velocities are specifically chosen for Greenland Ice Sheet and for illustration purposes only. For the global processing, including Arctic, Antarctic and all the other areas of the world, e.g. high mountain glaciers, we use the Copernicus DEM GLO-30 (https://spacedata.copernicus.eu/web/cscda/dataset-details?articleId=394198) and ITS_LIVE Version 1 Landsat-derived velocity mosaics (https://its-live.jpl.nasa.gov) as the reference velocities for deriving all our Version 2 ITS_LIVE products. For the Greenland and Antarctic ice sheets, ITS_LIVE Version 1 Landsat-derived velocity mosaics are combined with MEaSUREs Greenland Ice Sheet Velocity Map from InSAR Data V002 (Joughin et al., 2010; Joughin et al., 2015) and MEaSUREs InSAR-Based Antarctica Ice Velocity Map, Version 2 (Mouginot et al., 2012; Rignot et al., 2017), respectively.

2.1.2 ITS_LIVE Sentinel-1 Image-Pair Data Product

Authors should elaborate *offset tracking success* for readers. What criteria do they use for deciding offset tracking and do they calculate it quantitatively?

In response to the reviewer's comment, we have added the following clarification (line # 189-191)

*Here, the offset tracking at a given grid point is defined as "valid" if the determined pixel displacement using the current chip passes the NDC filter based on the similarity with adjacent chips, as further discussed in Section 2.2.*

"Hence, the selection of the appropriate sensor combination is dependent on the actual use case including data availability, quality, study area, etc." Such statements are made a couple of times without actually providing clear guidance on these criteria. The authors should clearly state how did they estimate such criteria for different regions on the globe. Authors should provide practical challenges in deciding these criteria while selecting input datasets.

We thank the reviewer on suggesting more technical details. We have removed such vague statements. Please refer to the major change #2 listed on the front cover of this response. In short, all of the sensor combination with time spans from 6 to 12 days were processed and stored in the data archive. Therefore, no criteria were imposed for generating the global products. However, from users' respective, recommendations have been made on how to select the appropriate data. In response to the reviewer's comment, an elaborated discussion on this has been added, where the relevant text (line # 196-208) is also copied here:

*Image pair products span the time period from the launch date of the corresponding sensors (e.g., Sentinel-1A—Sentinel-1A image pairs are available after the Sentinel-1A launch date, Sentinel-1B—Sentinel-1B and Sentinel-1A—Sentinel-1B or Sentinel-1B—Sentinel-1A image pairs are available after the launch date of Sentinel-1B) to present with a project goal of providing < 3 months latency from the date of acquisition. The separation times for the image pairs vary from 6 to 12 days for all sensor combinations, with the goal of extending to 60 days with multiples of 6-day separation dependent on funding. As a rule of thumb, for working with individual image-pair products over fast-flowing glacierized regions where rapid temporal decorrelation of radar signals is expected, using shorter separation times if available (e.g. 6 day; Sentinel-1A—Sentinel-1B or equivalently Sentinel-1B—Sentinel-1A) usually has better data quality than longer ones (e.g. 12 days or larger; Sentinel-1A—Sentinel-1A or Sentinel-1B—Sentinel-1B) due to less temporal decorrelation, which has been investigated by (Friedl et al., 2021; Solgaard et al., 2021). However, for observing slow-moving targets, all image-pair products with various separation times need to be considered to fully capture both the short-term and long-term variability of the velocities, such as in studying the seasonal ice dynamics (Greene et al., 2020).*

The output glacier velocity maps are generated in 120 m spatial resolution. Is this specifically because computational resources or authors have used a specific criterion for choosing this as the optimal resolution.

In response to the comment, we have added the following clarification (line # 165-167)

*For all map projections, a constant grid posting of 120 m is used to enhance the product resolution (by 50%) while keeping computational demands within reason. 240 m posting was used for Version 1.0 of the ITS_LIVE image pair products.*

How have authors calculated the magnitude of errors? The authors have discussed thoroughly geolocation and ionospheric errors introduced in the analysis and how they tried to overcome these errors. However, the final product error magnitude calculations seem to be missing. If I am not wrong, can you re-direct to the existing literature for elaborating this process?

Please refer to the major change #7 listed on the front cover of this response. In response to the reviewer's comment, we have elaborated the uncertainty calculation at the end of Section 3.1.2. Also, in the newly added Section 4 (cross-validation), we have provided the time series (thousands of epochs) results of the product uncertainty over three typical sites covering the globe.

"This sparse/dense combinative searching strategy substantially improve computational efficiency". I would like to see numbers on efficiency and improvement.

In response to the comment, we have revised the sentence as follows (line # 259-260)

> *This sparse/dense combinative searching strategy substantially improve computational efficiency (i.e. 2 orders of magnitude improvement; as further discussed in Section 2.4).*

General comments: Authors have demonstrated the processing on Greenland but state that the products are global. I am wondering if they can describe practical implementation challenges in processing global datasets and how uncertainties associated with other regions are dealt with. This is important because the research community will use these datasets for their research in the coming years and they would like to see practical challenges in the Arctic, Antarctic, mountain areas in the Himalayas, Alps. Also, the major limitation of this paper is that the authors have only focused on Greenland. I will suggest including a good distribution of test sites covering different parts of the globe.

Thanks to the reviewer on this comment. The newly added Section 4 includes the cross-validation with other similar products over three typical sites covering the globe: Jakobshavn Glacier in Greenland, Pine Island Glacier in Antarctic, and Malaspina Glacier in Alaska. With this detailed time series analysis of velocity / uncertainty, it is concluded that the algorithm and products are applicable to other regions of the globe besides Greenland Ice Sheet.

Validation: As we know that there are multiple velocity products are being generated both using SAR and optical methods. This study does not provide any guidance on comparing the present product with existing velocity products. This will be very useful for researchers as they would like to know reliable velocity products for their studies. Similarly, new algorithms are being developed for improvement in velocity products. This study does not comment on state-of-the-art methods and the comparison with other algorithms. Eventually, the scientific community will be benefited from such inter-comparison experiments.

Thanks to the reviewer very much for this constructive comment as well. In the newly added Section 4, we have also included the cross-validation with other similar products, e.g. PROMICE, FAU and other MEaSUREs as well as other ITS_LIVE Version 2 optical products (from Landsat-8 and Sentinel-2). Please also refer to the major change #1 listed on the front cover of this response.

Ground validation: Authors have not mentioned about validation of velocity products with in situ measurements. Have they attempted such validation?

Thanks for this comment. Indeed, currently we are not planning for in-situ data validation. As a similar ESSD-published product, PROMICE product (Solgaard et al., 2021) has been compared with in-situ GPS dataset because at the PROMICE project level, there are a number of in situ GPS measurements available from the PROMICE automatic weather stations (AWSs). However, for the global product validation, a large number of GPS stations would be required and thus need to be coordinated for matching the time and place of Sentinel-1 data acquisition. Therefore, in this work, we only cross-validated our ITS_LIVE Sentinel-1 products with other similar products (PROMICE, FAU and other MEaSUREs along with ITS_LIVE Landsat-8 and Sentinel-2) some of which may have been validated against in situ data (e.g. PROMICE). This treatment is also adopted by another similar ESSD-published product, FAU (Fiedl et al., 2021).

---

## Author Response (AR2)

**Handling Editor:**

I do not want to see you 'chasing' various mountain glaciers as requested by individual reviews but - at the same time - you will need to deal explicitly with issues of quality of velocity estimates for complex glaciers in difficult terrain.

Just an idea:

Choose mountain glacier(s) from among the list of (30?) reference glaciers at WGMS (https://wgms.ch/)? Certainly one to show optimal outcome of velocity techniques, but perhaps also one to show challenges?

Entirely your choice, you will know best how to present your work. Just ideas from this end.

We genuinely thank the Handling Editor for suggesting both ways above to present the data quality description over small mountain glaciers, which is an important part of the data validation. Since the manuscript is already very lengthy, we prefer to add a paragraph detailing all the possible limitations and errors that could be found in the dataset over high-relief small mountain glaciers. Please note that no single small mountain glacier can have all of these limitations, so the elaborated validation over various mountain glaciers (as the editor says "chasing various mountain glaciers") would be incomplete anyway. The newly added paragraph (line 750-761 in the revised text) is also copied here:

*"Below we provide the implications of the data quality over smaller mountain glaciers. For global processing, autoRIFT uses square search chip sizes ranging 240 m × 240 m to 1920 m × 1920 m. Only a single velocity vector is returned for a single search chip. This means that when a single search chip covers a surface with steep spatial gradients in surface velocity (e.g. shear margins, glacier margins, nunataks), only a single velocity vector will be returned. The returned vector represents the displacement between features that provide maximum correlation. Rock often dominates the correlation for mixed search chips that contain rock. This can cause glacier velocities to be negatively biased for narrow valley glacier and along shear margins. This same issue occurs for features that advect with the glacier (e.g. medial moraines) but present as stationary features. When a search chip samples these features the correlation can be dominated by the advecting moraine that appears as stationary. Lastly, Sentinel-1 is a side-looking SAR that is impacted by layover (i.e., multiple targets at the same range distance from the sensor are overlaid with each other causing their velocities mixed together) and line-of-sight shadowing (i.e., no targets appear at the side of a high mountain glacier facing away from radar resulting in missing velocity data) effects. Both of these issues are magnified in areas of high-relief where mountain glaciers are often located."*

**Reviewer #1:**

The revised manuscript is substantially improved and brings more clarity. Well done. I have two remaining concerns and a few minor corrections that may be quickly addressed in the final version.
1. Although the authors included a mountain glacier - Malaspina Glacier in the revised manuscript to show that their global dataset is good enough for mountain glaciers. To be honest, Malaspina is not a good representative of most of the mountain glaciers. Most mountain glaciers are small, even the largest mountain glacier from high-Asia is much smaller than Malaspina. It is quite clear that the ITS_LIVE velocities should be diligently used for these glaciers before any scientific use, mainly because of the spatial resolution. This must be addressed in the revised manuscript - A line in the Abstract and/or a few lines in the Conclusions and Outlook.
By the way, this was also raised by another reviewer.

Please refer to the response to the handling editor at the beginning of this response.

2. Another concern is for the scientific community who has to choose which among (e.g. FAU, ITS_LIVE, PROMICE) these data is the most appropriate (e.g. resolution, addressed uncertainty etc.). Therefore a table showing parameters such as Temporal and Spatial Resolutions, Period, Method, Uncertainty, Global/Regional etc. of these data would be a useful addition. I think the authors have already included them in validation, so adding a table like this won't be too much of a task.

We thank the reviewer very much for this constructive advice. We have included the new Table 1 in the revised text, with the screenshot attached below.

**Table 1.** Sentinel-1 based data product specifics of PROMICE, FAU and ITS_LIVE.

| Data Product | Temporal Resolution | Spatial Resolution | Period | Method | Uncertainty | Global/ Regional |
|---|---|---|---|---|---|---|
| PROMICE | 24 day (temporally averaged) | grid spacing 500 m (effective resolution: 800-900 m) | 2016-present | Normalized Cross Correlation | 20-27 m y$^{-1}$ (with in-situ GPS) and 8-12 m y$^{-1}$ (over stable ground) | Greenland only |
| FAU | 6-12 day (no temporal averaging) | grid spacing 200 m (effective resolution: 800-900 m) | 2016-present | Normalized Cross Correlation | 14.6 m y$^{-1}$ (with TerraSAR-X) and 87.6 m y$^{-1}$ (with Landsat-8) | Global |
| ITS_LIVE | 6-12 day (no temporal averaging) | grid spacing 120 m (effective resolution: 240-1920 m) | 2016-present | Normalized Cross Correlation | 50-70 m y$^{-1}$ (over rocks and stationary surfaces) | Global |

Other minor suggestions:

L235: One image can't be contaminated by temporal decorrelation.

Removed "one of". The revised sentence is

"if the input images were largely contaminated by temporal decorrelation"

L900: Unnecessary comma after Jakobshavn Is.,

Removed comma.

L915 or elsewhere: The texts on Qualitative comparisons are unnecessary and do not add value here - quite clear in the maps already. The time-series comparisons make more sense and this was nicely done.

Thanks to the reviewer for this suggestion. However, we prefer to keep some of these texts as some features in the figures need to be discussed to bring forward or induce the unique characteristics of our data product compared to the others. For example, when discussing Fig. 12, we mentioned "It can also be seen from Fig. 12 that the PROMICE product has been smoothed/filtered spatially as well as averaged temporally by mosaicking all the 6-day and 12-day image pair product within the 24-day temporal resolution window." Although readers can easily notice the strong filtering effect of PROMICE compared to ours, we wanted to emphasize to the readers and thus convey the idea that this phenomenon is because our data product has a much shorter temporal resolution of 6-12 days (without temporal averaging) compared to PROMICE's 24-day (with strong temporal averaging). We thus appreciate the reviewer's understanding that such texts are kept as they are.

Figure13: These comparative plots clearly show the strength of Sentinel-1 based velocities. You may indicate one of the periods when ITS_LIVE Optical data based velocities are not available in order to show this. Just an addition.

Per the reviewer's comment, we added the following sentence both in the text and Fig.13's caption:

"Note for each year of 2016-2022, there are periods that ITS_LIVE optical data are unavailable, which strengthens the adding value and competitive edge of using ITS_LIVE SAR data (currently Sentinel-1)."

L980: date >> data

Fixed

Figure 17: R2 >> R square

All $R^2$ in figure labels are marked as "R2", which was also used by other scientific articles. We appreciate the reviewer's understanding on this.

**Reviewer #3:**

The paper by Lei et al presents a processing methodology for Sentinel-1 imagery as part of the ITS_LIVE project, which mainly focused on Landsat data in the past. The paper presents in details the processing methodology of the processing chain, and SAR radar processing. My main comment is similar as one of the other reviewers, about the lack of application in mountainous areas. Here most of the applications are provided for Greenland, Antarctica and recently Alaska. Although ITS_LIVE claims to provide data on all glaciated massifs, all applications here are shown on large glaciers (Jakobshavn, Malaspina and Pine Island), with relatively smooth ice surfaces and relatively high flow velocities, compared to the great diversity of terrestrial glaciers. On top of that, satellite image processing in these regions is much more complex, with extremely high topography, shadow /layover problems, slower and smaller ice masses. Hence, monitoring the evolution of surface flow velocity in these areas is much more complex (see Millan et al., 2019), and specifically with synthetic aperture radar. Moreover, the choice of correlation parameters can vary largely depending on the ice object to be examined, which will also impact the spatial coverage and accuracy of the measurements. Thus, the use of velocity data to track glacier dynamics in mountain ranges can be highly risky when taking raw data as presented here. It is notably subject to advanced post-processing, with stacking or temporal interpolation (Millan et al., 2019; Charrier et al., 2021; Derkacheva et al., 2019), in order to extract a meaningful signal from noisy data. Since ITS_LIVE claims to have global coverage of velocities, I think it is important that the authors reconsider their test regions, to be more representative of the diversity of glaciers they want to cover. Thus I suggest keeping Jakobshavn in Greenland (a large and very fast glacier) but adding the Khumbu glacier region in the Himalayas, which is well representative of mountainous terrain issues, and the southern ice field of Patagonia, which has fast marine glaciers, but in areas of high topography and harsh weather conditions. I think it is also important that authors compare their composite map with existing products, in order to see the difference in velocity pattern, which is important for modelers but also for reconstructing ice volumes, or assessing rates of glacial erosions. For that matter I suggest to compare with the velocity data from Friedl et al., 2021, and Millan et al., 2022 and specifically in mountainous regions, where getting precise velocity is the most difficult.

We thank the reviewer for suggesting various test regions that are more focused on mountain glaciers. Rather than selecting a complete different set of test regions, please refer to our response to the handling editor at the beginning of this response for our addition on any possible problems of the dataset over small mountain glaciers. We appreciate the reviewer's understanding and clarify other concerns and/or points raised by the reviewer as below.

First, Millan et al. (2019) does not deal with radar data and we were unable to identify any specific approach presented in that paper that is relevant to the image-pair data being presented in our manuscript. Millan et al. (2019) does however provide a great discussion

on optical image correlation and data syntheses, two topics not covered in our study. Charrier et al., (2021) discusses a new method for post-processing of image-pair data, an important topic but not one covered in our study. We were unable to locate Derkacheva et al., (2019) but did find a Derkacheva et al., (2020) that discusses filtering approaches for post-processing image-pair data, a topic that we do not cover in our study. Our processing of the Sentinel-1 data can be viewed as an enhanced version of the approach taken by Friedl et al., (2021; https://essd.copernicus.org/articles/13/4653/2021/). In agreement with the findings presented in our study, Friedl et al. (2021) find that "Overall, we find that both [Landsat-8 and Sentinel-1 velocity] data sets are in good agreement. Absolute velocity differences are generally less than $0.02\,\mathrm{m\,d^{-1}}$ over stable ground and slow-moving ice with enough surface features that can be successfully tracked.". Lastly, we did cross-validated our data product with Friedl et al. (2021) as shown in our Fig. 20-21, as well as other similar products (PROMICE and other MEaSUREs along with ITS_LIVE Landsat-8 and Sentinel-2).

---

## Author Response (AR3)

**Handling Editor Comments to the author**:

We need the identical Zenodo link (e.g. from line 820) also at end of abstract (e.g. at line 31). We need to avoid any sense of divergent sources or versions.

We have revised the sentences to make sure they point to the single Zenodo link.

Manuscript contains many (too many?) residual spelling, typographical and labelling errors. E.g. if NSIDC provides the Arctic projections, who provides the Antarctic projections?

We were correct in calling the projections, i.e. there is "NSIDC" in front of Arctic projections while there is nothing to add to the Antarctic counterpart.

Or, in Figure 6, careful reader can discern which color represents autoRIFT and which ampcor but legend does not provide this information. (Reader finds better examples in Fig 8, Fig 9.)

Both titles of Fig. 6's subfigures have been revised to remove the confusion.

Line 582, velocity of Jakobshavn: statement about speed assumes we know accurate velocities for all other (major?) glaciers? At line 646 reader encounters another 'fastest' (Pine Island) glacier, thinning in this case? (I think I would have a bit more caution.)

Two citations have been added to justify Jakobshavn as the fastest glacier in the world. We removed the word "fastest thinning" for Pine Island (although it can also be justified by the newly added citations), and call it "one of the fastest glaciers and rapid thinning", where two other citations have been added to justify it.

Furthermore, we added one more citation to Malaspina glacier regarding the Editor's concern.

Need better labelling in legend for Figure 12?

All of the legends in the figures that are spatial maps have been revised to have better labeling.

Most remaining errors should get identified and fixed during type-setting and proof-reading. Please can I suggest that authors recruit a colleague familiar with satellite SAR processing to serve as additional reader of proof copy? You (and we) really want highest quality description to accompany this important product.

We have asked one of our colleagues who is a radar expert to double check the radar terminology use and made sure the statements are appropriate and correct. We also made several minor fixes in the final proofreading.

Finally, thanks to the Editor for the constructive advices for improving and advertising the work, as well as the favorable consideration.

Yang Lei

On behalf of the authors